# Sampling Error in Aircraft Flux Measurements Based on a High-Resolution Large Eddy Simulation of the Marine Boundary Layer

Grant W. Petty[1]

[1]Atmospheric and Oceanic Sciences, University of Wisconsin-Madison, 1225 W. Dayton St, Madison, WI, 53706, USA

**Correspondence:** Grant W. Petty (gwpetty@wisc.edu)

**Abstract.** A high-resolution (1.25 m) LES simulation of the nocturnal cloud-topped marine boundary layer is used to evaluate random error as a function of continuous track length $L$ for virtual aircraft measurements of turbulent fluxes of sensible heat, latent heat, and horizontal momentum. Results are compared with the widely used formula of Lenschow and Stankov (1986). In support of these comparisons, the relevant integral length scales and correlations are evaluated and documented. It is shown that for heights up to approximately 100 m ($z/z_i = 0.12$), the length scales are accurately predicted by empirical expressions of the form $I_f = Az^b$. The Lenschow and Stankov expression is found to be remarkably accurate at predicting the random error for shorter (7–10 km) flight tracks, but the empirically determined errors decay more rapidly with $L$ than the $L^{-1/2}$ relationship predicted from theory. Consistent with earlier findings, required track lengths to obtain useful precision increase sharply with altitude. In addition, an examination is undertaken of the role of uncertainties in empirically determined integral length scales and correlations in flux uncertainties as well as of the flux errors associated with crosswind and along-wind flight tracks. It is found that for 7.2 km flight tracks, flux errors are improved by factor of approximately 1.5 to 2 for most variables by making measurements in the crosswind direction.

## 1 Introduction

### 1.1 Background

The eddy covariance (EC) method has long been the principal means of making field measurements of turbulent fluxes of energy, momentum, and trace gases in the planetary boundary layer (PBL). While towers are commonly used to measure fluxes over longer time periods (weeks to years) at fixed locations, aircraft are the preferred platform for obtaining estimates of fluxes over larger areas, albeit for far shorter periods of time.

By their nature, EC estimates of fluxes are averages of quantities that randomly fluctuate on time scales ranging from fractions of a second to hours and over distance scales from centimeters to kilometers. Thus, in addition to the usual factors

affecting measurement quality from any instrument, such as calibration, precision, time response, and site representativeness, the quality of eddy correlation estimates are critically subject to statistical sampling error as well as to potentially restrictive assumptions about temporal stationarity and/or spatial homogeneity.

In this paper, we are concerned exclusively with the problem of random sampling error in the context of aircraft flux measurements relying on in situ (e.g., gust probe) measurements of fluctuating wind vector and scalar quantities directly along the flight path. The key question that has been asked for decades is, how long of a flight path — whether consisting of a single extended leg or a series of shorter legs over a smaller region — is sufficient to obtain turbulent flux estimates of useful precision? This problem was examined by Lenschow and Stankov (1986), Lenschow et al. (1994), and Mann and Lenschow (1994),

relying on statistical models of turbulence supported by observations. Mahrt (1998) further examined the sampling problem in the context of the problems posed by non-stationarity. Others took an empirical approach, comparing aircraft flux estimates with those from nearby fixed towers (Desjardins et al., 1989; Grossman, 1992; Mahrt, 1998).

Finkelstein and Sims (2001) used what was characterized as a more general yet rigorous derivation to arrive at a different expression for the flux sampling error. Based on the properties of test data sets for several variables, their computed flux errors

were typically about a factor of two larger than those predicted by the expressions of Mann and Lenschow (1994).

In recent years, it has become possible to use large eddy simulation (LES) models to explicitly resolve turbulent fluctuations of mass, energy, and motion in the planetary boundary layer at fairly fine scales, leaving only a small fraction of the total turbulent exchange to subgrid-scale parameterizations, especially at levels much above the surface. Model turbulence fields can be directly compared with aircraft measurements in the same environment, as was done for example by Brilouet et al.

(2020) for a cold-air outbreak over the northwest Mediterranean, who found that the LES successfully reproduced convective structures observed by the aircraft.

Alternatively, the LES fields may serve as a virtual environment within which turbulence may be sampled in a manner consistent with the platform (Schröter et al., 2000; Sühring and Raasch, 2013; Sühring et al., 2019). An attractive feature of this second approach, which is the focus of this paper, is the unique ability to compare the flux estimate obtained from a limited

sample with the known "true" domain-averaged value, in contrast to the case for real-world comparisons between inherently disparate aircraft and tower measurements of the area-averaged flux.

A closer look at Schröter et al. (2000) is worthwhile in light of significant similarities in both their objectives and their approach to those of the present paper. They flew virtual aircraft through a $401\times401\times42$ LES domain simulating a continental convective boundary layer at heights ranging from 175 m to 1075 m. "Measured" sensible heat fluxes were compared with

"true" domain-averaged fluxes, and the ensemble flux errors determined. These errors were found to compare well with expressions of Lenschow and Stankov (1986) discussed below. They found that 2,000 sec flight durations, corresponding to a distance of 200 km, were sufficient to achieve 10% precision in the flux estimates.

The more recent studies by Sühring and Raasch (2013) and Sühring et al. (2019) have focused on using virtual aircraft flights through LES domains to assess the ability to resolve the effects of surface inhomogeneities on aircraft flux measurements.

An important limitation of LES studies has been the tradeoff between domain size and the ability to resolve turbulence at the finest scales, a consequence of the high computational cost of combining large domains and short time steps. For this reason,

even fairly recently published studies using LES to study the flux sampling problem have utilized horizontal grid dimensions of order $2 \times 10^3$ or less and horizontal grid resolutions of 7–50 m (Table 1). Coarser grid resolutions imply a significant role for parameterized subgrid-scale fluxes and preclude a complete examination of the flux sampling problem for low- and slow-flying light or ultralight aircraft (Metzger et al., 2011; Vellinga et al., 2013) and unmanned aerial vehicles (UAVs; Elston et al. 2015).

The present study is motivated by the recent availability of LES results for a 4096×4096 domain with 1.25 m solution, offering an unusually large range in resolvable scale. While the resulting 5.1 km domain size is still insufficient to capture some potentially important larger-scale modes of variability (e.g., Brooks and Rogers 1997; Brilouet et al. 2017), it is a sufficient improvement to motivate a fresh look at the aircraft flux sampling problem. This is especially true when the focus is expressly restricted to relatively low-level, purely turbulent motions having scales comparable to or less than the depth of the boundary layer.

## 1.2 Objectives

The central purpose of this paper is to investigate the empirical relationship between sampling error in turbulent flux measurements and the length of a continuous, ideal aircraft track through the virtual atmosphere represented by an LES. Specifically, the analysis is based on a single time step of the LES run described by Matheou (2018). Because the simulation utilized cyclic lateral boundary conditions, it is possible to construct ensembles of simulated flight paths that are continuous over longer distances and yet do not resample the same locations in the domain. We are thus able to focus narrowly on the precision of aircraft flux measurements as a function of flight track length alone in the particular environment represented by this simulation without the complication of boundary effects or the averaging of shorter segments.

To put the empirical determinations of error into perspective, results are compared with the widely used formula of Lenschow and Stankov (1986). That relationship in turn depends on estimates of the so-called integral length of the fields being measured, so documenting integral lengths obtained from the LES fields and their dependence on height and wind direction relative to flight tracks is a related objective. Additionally, the relative error of crosswind versus along-wind flux measurements is examined for a single track length of 7.2 km.

As previously noted, the general methods and motivations are in many ways similar to those of Schröter et al. (2000) and Sühring and Raasch (2013). However, horizontal inhomogeneity of the surface and thus of the boundary layer forcing is eliminated as a factor in the present analysis, domain resolution is improved allowing the simulation of lower-level flights (down to 10 m), and the environment considered here is a midlatitude summertime marine boundary layer topped by a stratocumulus cloud deck.

## 1.3 Overview

In the following section, I describe the LES setup and the simulated environment, and I present selected features of the simulation results that lend confidence in the apparent realism of the LES as it relates to the present analysis. Section 3 describes the calculation of turbulent flux quantities from the LES fields and presents ogive plots depicting the dominant spectral contributions to vertical fluxes at selected heights about the ocean surface. Section 4 introduces the problem of characterizing random

error in flux measurements from finite aircraft tracks and focuses in particular on the determination of integral length scales and correlations for turbulent quantities required by widely used expressions for the flux sampling error. In Section 5, I describe the algorithm for defining continuous, periodic flight tracks of varying lengths and for obtaining "virtual" flux measurements along those tracks. Section 6 describes key results of the analysis, including the following components: 1) empirical determinations of sampling errors as a function of track length, 2) comparisons with the widely used expressions of Lenschow and Stankov (1986), 3) an empirical examination of the potential contribution of uncertainties in estimates of integral lengths and correlations from flight tracks to the total random error, and 4) a comparison of flux errors for crosswind and parallel tracks for 7.2 km tracks. A short summary of key results is presented in Section 7.

## 2    Data

### 2.1    LES description

In one of the largest LES model runs ever published, Matheou (2018) used a buoyancy-adjusted stretched-vortex model (Chung and Matheou, 2014; Matheou and Chung, 2014) to simulate a nocturnal stratocumulus case over the Pacific ocean southwest of Los Angeles (vicinity of 32°N, 122°W) corresponding to the 10 July 2001 research flight (RF01) of the second Dynamics and Chemistry of Marine Stratocumulus (DYCOMS-II) field study (Stevens et al., 2003).

The LES grid resolution is 1.25 m in $x$, $y$, and $z$, and the grid size is 4096×4096×1200, corresponding to a 5.12×5.12×1.5 km domain with cyclic lateral boundary conditions. Because of the considerable computing resources required, it was only possible to simulate the atmospheric boundary layer for 2 h of physical time. Nevertheless, the simulation required 36 days of wall-clock time on 4,096 CPU cores at the NASA Advanced Supercomputing (NAS) Division at Ames Research Center.

The single final time step of the simulation, comprising a total of ∼800 GB of gridded data, was utilized for this study. Output included all grid-resolved model variables, including temperature $T$, specific humidity $q$, wind velocity components $u$, $v$, and $w$, and cloud water content $q_l$.

### 2.2    Simulated environment

The environment was characteristic of the summertime closed-cell marine stratocumulus that prevails over the ocean off the southern California coast. The operations summary for the flight day describes the case as "very homogeneous" (Earth Observing Laboratory, 2007).

Consistent with the observed environment on the flight date, the LES was initialized with a boundary layer depth $z_i =$840 m delineated by a sharp 8.5 K temperature inversion. Below $z_i$, the initial potential temperature $\theta$ was 289 K and the specific humidity $q$ was 9 g kg$^{-1}$. These values did not change significantly during the simulated evolution of the boundary layer (Fig. 1). A prescribed sea surface temperature of 292.5 K was used, leading to domain-averaged sensible and latent heat fluxes of 15 W m$^{-2}$ and 115 W m$^{-2}$, respectively. The wind was initialized as geostrophic; at the conclusion of the simulation,

the mean horizontal wind components $\overline{u}$ and $\overline{v}$ below $z_i$ were nearly uniform at 6.7 m s$^{-1}$ and $-5.0$ m s$^{-1}$, respectively, corresponding to a wind direction of 307° (approximately northwesterly) and speed of 8.4 m s$^{-1}$.

The cloud base and top, defined here as the lowest and highest levels at which non-zero cloud liquid water occurred at any point in the horizontal domain, were found at 500 m and 885 m, respectively. This paper is concerned exclusively with the sub-cloud boundary layer at and below 400 m or $0.48z_i$.

## 2.3 Turbulent structure and domain-averaged fluxes

Many details concerning the model run and the overall turbulent structure of the simulated boundary layer are given by Matheou (2018). Here we focus on those grid-resolvable turbulent properties most directly relevant to an understanding of the simulated flux measurements.

Throughout this paper, results are highlighted at four representative heights: 10 m, 40 m, 100 m, and 400 m. For the environment simulated, these correspond to approximately $0.01z_i$, $0.05z_i$, $0.12z_i$, and $0.48z_i$, respectively. While subjectively chosen, these heights have some correspondence to real-world aircraft observations, including Cook and Renfrew (2015), who observed marine fluxes around the British Isles at an altitude of $\sim$40 m, and the recent Chequamegon Heterogeneous Ecosystem Energy-balance Study Enabled by a High-density Extensive Array of Detectors 2019 campaign (CHEESEHEAD19), in which the University of Wyoming King Air flew legs at 100 m and 400 m (Butterworth et al., 2021). The lowest height of 10 m was included because of interest by author in potential flux measurements using the University of Wisconsin research ultralight airplane.

### 2.3.1 Spectra

Power spectra were computed for vertical velocity $w$, temperature $T$, specific humidity $q$, and horizontal wind speed $U$ for each of above levels. This was done by first taking the 2-D Fourier power spectrum of the entire horizontal slice through the domain and then averaging radially to obtain a direction-independent 1-D average power spectrum. For clarity, results are shown for only two heights, 40 m and 400 m, in Fig. 2.

Figure 2a shows that, at 400 m height ($0.48z_i$), the LES produces a vertical velocity spectrum approximately obeying a Kolmogorov power law for the vertical velocity field over more than two decades, from wavelengths greater than 1 km down to $\sim$8 m or about three times the Nyquist wavelength of 2.5 m. At this height, peak energy in the spectrum is near 1 km wavelength. At the much lower height of 40 m ($0.05z_i$), peak energy is found near 150 m, and there is substantially less energy at longer wavelengths. In both cases, the peak energy in $w$ is thus found near $\sim$4 times the physical distance $z$ from the lower boundary, consistent with dominant contributions from eddies of size $z$ centered at height $z$.

The spectra for temperature (Fig. 2b) and specific humidity (Fig. 2c) reveal considerably more energy at the longest wavelengths, even at 40 m height, while also rolling off earlier than $w$ at the short wavelength end. The spectrum for temperature is also somewhat flatter than for either $w$ or $q$. The spectrum for horizontal wind speed $U$ (Fig. 2d) is notable in having no significant dependence on height, presumably because proximity to the lower boundary imposes no special constraint on horizontal flow in this low-friction, neutrally stratified environment.

Overall, the spectra of scalar variables seem reasonable and give no clear cause to doubt the overall realism of the LES for wavelengths spanning the range from $\sim$10 m to $\sim$1 km. Nevertheless, the constraint on the long-wavelength end due to the approximately 5 km domain size cannot be overlooked. While it is easy to discount the importance of the very short wavelengths that are artificially dissipated in the numerical model, it is more difficult to assess the influence of the domain size on turbulent structures of order 1 km and greater. de Roode et al. (2004) found that peak energy for vertical motion and for virtual potential temperature are found at wavelengths comparable to the boundary layer depth—which is less than one kilometer in the present case—but that humidity and potential temperature can exhibit significant variations on larger scales.

### 2.3.2  Horizontal cross-sections

Horizontal cross-sections of temperature $T'$, specific humidity $q'$, and vertical velocity $w'$ are presented for these heights in Figs. 3, where the primed quantities indicate instantaneous, local deviations from the domain-averaged value at each height. These show an evolution from predominantly fine-scale structure near the surface to consolidated perturbations of order 1 km diameter at 400 m. In the $w'$ field near the surface, there is a visually obvious alignment of structures with the mean wind. This directional anisotropy is far less apparent in $q'$ and is completely absent from $T'$ at low levels.

## 3  Flux Calculations

The local contributions to the turbulent fluxes of sensible SH and latent heat LH are

$$\mathrm{SH}' = \overline{\rho} C_{\mathrm{p}} \theta' w' \tag{1}$$

and

$$\mathrm{LH}' = \overline{\rho} L q' w', \tag{2}$$

where $\overline{\rho}$ is the mean air density at level $z$, $C_{\mathrm{p}} = 1005$ J kg$^{-1}$ K$^{-1}$ is the specific heat capacity at constant pressure, $\theta'$ is the potential temperature perturbation, and $L \approx 2.5 \times 10^6$ J kg$^{-1}$ is the latent heat of vaporization of water. Note that for the low altitudes and near-standard pressures $p$ encountered in this simulation, $\theta' = T'(100 \,\mathrm{kPa}/p)^{0.286} \approx T'$.

In the above expressions, the common notation $LE$ for latent heat flux is avoided because it implies a product of the latent heat of vaporization $L$ and surface evaporation rate $E$. At any significant distance above the surface, the actual vertical flux of water vapor measured by an aircraft may or may not equal $E$. All fluxes described herein should be understood as flight-level fluxes without any assumption about their relationship to surface fluxes. To reinforce this distinction, the more generic symbols SH and LH have been adopted here.

The local $u$ and $v$ components of momentum flux are computed as

$$(\tau_u', \tau_v') = \rho \cdot (u'w', v'w'). \tag{3}$$

Where appropriate in the discussion below, the horizontal wind vector $(u, v)$ is rotated into a coordinate system aligned with the mean wind direction so as to isolate the along-wind and crosswind components $(U_\parallel, U_\perp)$.

Fig. 4 depicts the ratio of parameterized subgrid-scale flux to total vertical flux (subgrid-scale plus resolved) as a function of height above the surface. At 10 m height, the fraction is less than 5% for LH and SH and less than 2% for horizontal momentum flux. Consequently, we can neglect the unresolved components of the fluxes for the purposes of this study.

Domain-averaged resolved fluxes and turbulent kinetic energy (TKE) are depicted in Fig. 5. These fluxes serve as the nominal "truth" values against which simulated aircraft flux measurements will be evaluated.

As expected, the profile of sensible heat flux (Fig. 5a) is quite linear except within cloud, where latent heating due to condensation sharply increases the positive correlation between $T$ and $w$. It crosses through zero near 445 m ($0.53z_i$). The profile of latent heat flux is noticeably less smooth, as are $\tau_u$ and $\tau_v$ (Fig. 5b). Turbulent kinetic energy varies only weakly below cloud and has a sharp maximum at cloud top, as expected. (Fig. 5c).

Figure 6 presents ogive (cumulative distribution) plots of the spectral contributions to each flux. For all four flux variables, there is a pronounced shift from predominantly short-wavelength contributions at the lowest levels to much lower-frequency contributions at 400 m. In particular, 90% of the total sensible heat flux contribution comes from the wavelength ranges [6 m, 146 m], [8 m, 0.51 km], [12 m, 1.0 km], and [17 m, 2.6 km] at heights of 10 m, 40 m, 100 m, and 400 m, respectively, with qualitatively similar ranges for latent heat flux and momentum flux. Overall, these results suggest that flux error assessments for heights near or below 100 m are least likely to be seriously biased by the LES domain size of 5.12 km, while 400 km may be more susceptible.

## 4 Random Error in Flux Measurements

### 4.1 Defining the problem

According to the eddy covariance method, an estimate $\hat{F}_\psi$ of the true turbulent flux $F_\psi$ is obtained from a series of closely-spaced measurements of scalar variable $\psi$ and vertical velocity $w$ as

$$\hat{F}_\psi \propto \overline{\psi'w'}, \tag{4}$$

where the proportionality allows for appropriate multiplicative constants, and the overline indicates an average over time and/or space, with $\psi' \equiv \psi - \overline{\psi}$ and $w' \equiv w - \overline{w}$. In the present context, the average is along the length of a flight track and is considered to be effectively instantaneous.

Because of the stochastic nature of turbulence, $\hat{F}_\psi$ is in general an imperfect estimate of the true flux $F_\psi$, but one that can theoretically improve with a larger sample, subject to assumptions about stationarity. For a single flight track, the error in the flux estimate is

$$\delta_f \equiv \hat{F}_\psi - F_\psi. \tag{5}$$

If a large ensemble of identical tracks were flown but sufficiently displaced from one another in time and/or space so as to be statistically independent, the mean error would be expected to converge to zero, but the uncertainty associated with any single

track could be characterized via the ensemble root-mean-squared error (Lenschow and Stankov 1986, Eq. 1):

$$\sigma_\psi \equiv \langle (\hat{F}_\psi - F_\psi)^2 \rangle^{1/2}. \tag{6}$$

The fundamental problem is to characterize the above random uncertainty for any particular flight track, given the length of the track and appropriate assumptions about the turbulent environment being measured. An additional complicating factor is that the required true means $\overline{\psi}$ and $\overline{w}$ must themselves be estimated from the finite flight track and will therefore also be subject to random errors.

## 4.2 Integral length scales

### 4.2.1 Significance

Central to the problem of estimating random error in aircraft measurements of turbulent fluxes is the integral length scale, also known as the autocorrelation length, of the fields being measured (Lenschow and Stankov, 1986; Lenschow et al., 1994; Sühring and Raasch, 2013):

$$I_f \equiv \int_0^{r_0} A_f(r)\, dr, \tag{7}$$

where $A_f(r)$ is the autocorrelation of generic spatial function $f(x, y)$ at displacement $r = \sqrt{x^2 + y^2}$ in any specified direction, $A_f(0) \equiv 1$, and $r_0$ is the distance to the first zero crossing or one-half of the domain diameter (in this case 2.56 km), whichever is smaller. In the ideal "red noise" case, $A_f(r) = \exp(-r/I_f)$, and $A_f(I_f) = e^{-1}$. The length scale is thus useful for characterizing the minimum track length and/or spatial separation required in order to be able to treat two sets of measurements as statistically independent.

Viewed simplistically, for a track length $L = NI_f$, the random error in $\hat{F}_\psi$ as an estimate of the true ensemble value should decrease with $N^{-1/2}$, consistent with the calculation of standard error for any set of $N$ noisy measurements. Richardson et al. (2006) argues persuasively that it is more natural to focus on the absolute rather than relative random error, as the relative error becomes extremely large or even undefined in the common case that that the flux approaches or crosses zero; for example, either at sunrise or sunset or near the altitude at which sensible heat flux switches signs. With that in mind, equation (7) of of Lenschow and Stankov (1986) is rearranged to give the following absolute random uncertainty:

$$\sigma_\psi = \left[ 2(\rho_{w,\psi}^{-2} + 1) \frac{I_{w\psi}}{L} \right]^{1/2} |F_\psi| \tag{8}$$

where $\rho_{w,\psi}$ is the zero-lag Pearson correlation coefficient of $w'$ with $\psi'$, and $I_{w\psi}$ is the integral scale length of $w'\psi'$.

Lenschow et al. (1994) and Mann and Lenschow (1994) refer to the difficulty of obtaining estimates of $I_{w\psi}$ and propose theoretically derived substitutions based on the more commonly available $I_w$ and $I_\psi$. The problem of convergence of integral lengths under some circumstances is further discussed by Durand et al. (2000). Subsequent authors, citing Mann and Lenschow (1994), variously utilize

$$I_{w\psi} \leq \sqrt{I_w I_\psi} \qquad ; \qquad \text{e.g., Sühring and Raasch (2013)} \tag{9}$$

or

$$I_{w\psi} \leq \frac{\sqrt{I_w I_\psi}}{\rho_{w,\psi}} \qquad ; \qquad \text{e.g., Bange et al. (2002).} \tag{10}$$

Because all of the above length scales and correlations can be computed directly from the LES fields, it will be possible to empirically assess the validity of (8)–(10).

### 4.2.2 Determination from LES

For each height under consideration, the two-dimensional autocorrelation of the full horizontal model domain was obtained for each turbulent field of interest. $I_f$ was then determined in two orthogonal directions: one aligned with the mean wind ("parallel") and the other perpendicular ("crosswind"). Results are shown in Fig. 7. Note that for momentum fluxes, the wind vector $(u, v)$ was projected onto the parallel and crosswind directions to obtain the rotated vector $(U_\parallel, U_\perp)$. Thus, the integral length scale labeled "$U_\perp$ Parallel", for example, describes that computed for the turbulent flux of the momentum component perpendicular to mean wind measured along a track parallel to the mean wind.

A number of general observations can be made concerning the computed values of $I_f$:

– For every variable, $I_f$ at and below $z = 100\,\text{m} = 0.12 z_i$, corresponding roughly to the surface layer, is well-approximated by a straight line on a log-log plot, implying an empirical relationship of the form $I_f \approx A z^b$. For some variables, such as the specific humidity (Fig. 7a) and the sensible and latent heat fluxes (Figs. 7f,g), the validity of such a relationship seems to extend considerably higher.

– For most variables, $I_f$ is larger for tracks parallel to the mean wind, as expected, than for crosswind tracks, often
considerably larger. Two interesting exceptions are found in $I_f$ for temperature above 40 m (Fig. 7b) and for $U_\parallel$ above 100 m, where the crosswind value of $I_f$ is up to 50% greater than for the parallel direction.

– For many variables ($q$, $U_\perp$, $w$, SH, and LH), the difference between the parallel and crosswind values of $I_f$ is largest near the surface and decreases to near zero at higher altitudes. For temperature, however, the reverse is true (Fig. 7b).

– The three turbulent wind components behave quite differently, with $I_f$ for $U_\parallel$ and $U_\perp$ exhibiting a relatively weak and
non-linear dependence on height, while for the vertical wind component $w$, the dependence is strong. Indeed for the crosswind case, $I_f$ for $w$ is almost perfectly proportional to $z$.

– Of all the variables considered, only the turbulent kinetic energy exhibits an integral length scale that is virtually constant with height throughout the full depth of the model domain (Fig. 7i).

The calculated $I_f$ results below 100 m were used to obtain coefficients $A$ and $b$ for the aforementioned power-law fit. These
coefficients are provided in Table 2. Alternatively, $I_f = A_\star z_\star^b$, where $z_\star \equiv z/z_i$, $A_\star = A z_i^b$ and $z_i = 840\,\text{m}$. It is not known to what degree the results found here are transferable to other environments.

It is noteworthy that the exponents $b$ found here for fluxes of sensible heat, latent heat, and horizontal momentum are quite different than the exponent of $-1/3$ or $-1/2$ obtained theoretically by Lenschow and Stankov (1986) for all of these (their

equation 11), being closer to $-1$ for the crosswind length scales and $-2/3$ for the along-wind length scales. It is beyond the scope of this paper to investigate the reasons for the discrepancy.

Apparent in the above results is that (9) and (10) generally yield large overestimates of $I_{w\psi}$ for any scalar variable $\psi$. For example, at 100 m height and measured in the crosswind direction, $I_q = 220$ m and $I_w = 73$ m. $\sqrt{I_q I_w}$ is then 127 m, a factor of 2.5 larger than the directly computed $I_{wq} = 51$ m. In general, the minimum overestimate at or below 400 m was found to be about a factor of 2 or greater, with ratios in excess of 3–5 typical at lower heights. The difference increases further if one uses the correlation coefficient $\rho_{w,\psi} < 1$ in the denominator per (10). Specific combinations of variables and heights can be easily examined using the relationships in Table 2.

Also required in the evaluation of random flux errors from (8) is the correlation $\rho_{w,\psi}$ between the vertical velocity $w'$ and the scalar variable $\psi'$ of interest. Computed profiles of $\rho_{w,\psi}$ are given in Fig. 8. For horizontal momentum, $\rho_{w,U} = \sqrt{\rho_{w,u}^2 + \rho_{w,v}^2}$.

## 5 Simulated aircraft measurements

### 5.1 Track definitions

As previously noted, the central purpose of this paper is to investigate the empirical relationship between sampling error in turbulent flux measurements and the length of a continuous, ideal aircraft track through the virtual atmosphere represented by the LES domain. The algorithm for defining these tracks is described below.

First, each track is required to be perfectly cyclic, albeit with a period $L$ substantially longer than the dimensions of the domain. Specifically, track angles relative the $x$-axis are defined by $\phi_n = \arctan(n)$, where $n$ is a positive integer. For $\phi_1 = 45°$, the track makes a single pass through the domain before returning to its starting point, and the track length is 7.2 km. A continuation of the track would simply resample the same grid points. For $\phi_2 = 63.4°$, the track makes two widely separated passes through the domain before returning to its starting point, and the non-repeating track length is 11.4 km. The maximum value of $n$ used was 20, for a 102.5 km continuous, non-overlapping path.

For any given $n$, starting points for different tracks of the same length are uniformly distributed along the boundary of the domain, maintaining prescribed minimum separation (measured perpendicular to the paths) intended to ensure a reasonable degree of statistical independence. These minimum separations were chosen to be greater than $I_f$ for all relevant variables at the level in question; 20 m, 50 m, 100 m, and 150 m were assumed for heights of 10 m, 40 m, 100 m, and 400 m, respectively.

The effect of the minimum separation requirement is to achieve a roughly constant overall sampling density for the ensemble of tracks of a given length. Thus, $N_n$ is small for long tracks and larger for short tracks. By taking mirror images and 90° rotations of the tracks defined in this way, the total number of distinct tracks available is actually $4N_n$, except for $n = 1$, in which case it is $2N_n$ owing to duplication when rotating and flipping the 45° track patterns. Figure 9 depicts an example track for $n = 3$ (top) and the full corresponding set of 32 distinct tracks (bottom) for a relatively wide minimum separation of 200 m used for visual clarity; the number of distinct tracks is significantly greater in the actual analysis.

Note that the algorithm described above is more elaborate than the fixed-angle track definitions utilized by Schröter et al. (2000) and Sühring et al. (2019). To reiterate, each track of length $L_i$ is required to be cyclic with period $L_i$, and individual

realizations of a given track length are distributed so as to both maximize and equalize their separation, subject to the afore-mentioned minimum separations. This allows us to obtain an optimal set of quasi-independent tracks. Constraining the tracks to be cyclic also means that there is by definition no trend or unsampled low frequency contribution to be concerned about—"measured" fluxes are the same regardless of the starting point along the track, as long as the total distance followed by the virtual aircraft is $L_i$.

An unavoidable shortcoming of the method used here is that track orientations relative to the mean wind cannot be specified separately, as the absolute orientations are uniquely determined by the chosen number of passes $n$ though the domain and thus by the track length. As $n$ increases, the trend is toward tracks that are oriented mostly north-south or east-west, whereas the mean wind in this simulation is from the northwest. Only for $n = 1$ are the track directions nearly aligned with or perpendicular to the wind direction.

## 5.2 Virtual flux observations

The "truth" value for each comparison is the domain-averaged flux value computed at each level (Fig. 5). These values utilized the deviations (primed quantities) from the true ensemble mean values for the entire domain. As measured against the true values, aircraft measurements are subject to two statistical sources of error: 1) error in the determination of the mean values relative to which primed quantities will be calculated , and 2) insufficient sampling of the fluctuating primed quantities themselves. Our virtual measurements simulate both effects by utilizing the track-determined mean values rather than the domain-averaged values. Both sources of error decrease asymptotically to zero in the limit of an infinitely long flight track, and the flux estimate will converge on the true ensemble value.

In the finite 5.12 km×5.12 km horizontal domain of our LES, there is the additional issue that a single long flight track can effectively sample the entire domain, so that the aircraft measurement can converge almost perfectly on the "true" value from measurements along a long but finite path. In principle, then, the results obtained herein for long flight tracks should be considered lower bounds on the expected random error. However, because of the minimum separation maintained between parallel segments of a given tracks, we believe this potential bias is unimportant for the flight track lengths considered here. More important may be the inability of the finite LES domain to reproduce flux variations with wavelengths greater than the domain size. As previously discussed for Fig. 6, this seems likely to be an issue mainly at heights of ∼400 m and above.

Finally, a notable constraint rooted in the availability of only a single time step from the LES is that the virtual aircraft effectively traverses each path instantaneously. We are in effect relying on Taylor's "frozen turbulence" hypothesis and, following Lenschow and Stankov (1986), postulate that this hypothesis is valid in the present case if $I_f < VT_f$, where $I_f$ is the integral length scale, $V$ is the speed of the aircraft, and $T_f$ is the autocorrelation time of the flux field being measured. Unfortunately, it is not possible to determine $T_f$ from the single time-step available for this study, but for a real-world $V \approx 85$ m sec$^{-1}$ similar to that of the Univerity of Wyoming KingAir, one would only require $T_f > 1$ s to satisfy the above inequality for flux measurements at a height of 100 m, based on the values of $I_f$ seen in Fig. 7. For a much slower aircraft or UAV with an airspeed of 20 m sec$^{-1}$, the requirement for $T_f$ is proportionally longer, but these have the option of flying at lower altitudes where $I_f$ is smaller.

## 6 Results

Raw sample results are presented in Fig. 10, with each blue dot representing the estimated flux from a single flight track of the indicated length. The true domain-averaged values are indicated as dashed black lines, and the root-mean-square (RMS) deviation from the true value is depicted as a red dashed line. For all flux variables, there is the expected decay in sampling error as the track length becomes longer.

Noteworthy features include the following:

- Overall scatter is largest at the highest altitudes, consistent with the longer integral length scales $I_f$ and with the observation by previous authors that error tends to increase with height (e.g., Grossman 1992).

- For sensible heat flux at 400 m (Fig. 10j), the magnitude of the scatter is often far in excess of the very low (1.8 W m$^{-2}$) mean flux itself, so that shorter track lengths cannot reliably determine even the sign of the mean flux.

- Relative to the mean values, scatter is particularly large for momentum flux. Because this quantity is positive-definite ($\tau = \sqrt{\tau_u^2 + \tau_v^2}$), the distribution of errors at 400 m is strongly skewed and positively biased.

Fig. 11 offers a closer look at the dependence of sampling error on path length and allows the results obtained to be compared with the random error predicted by (8). The red dots correspond to the empirical RMS error values previously represented in Fig. 10 as red dashed curves. The black dashed lines represent fits to those values. The accompanying power-law expressions for each fit appears in black as well as being summarized in Table 3.

The blue lines represent (8), where both $I_f$ and the correlation coefficients $\rho_{w,\psi}$ are computed directly from the 2-D LES-generated fields at each level and represent averages of the crosswind and parallel values. As previously pointed out, most simulated flight tracks are neither parallel nor perpendicular to the mean flow, and the set of available orientations varies with track length. For both reasons, no single representation of (8) as a curve in Fig. 11 can capture the variability of $I_f$ and $\rho_{w,\psi}$. However, especially for longer path lengths, most paths cross the mean wind at an oblique angle; thus the use of the average (direction-independent) $I_f$ and $\rho_{w,\psi}$ seems reasonable for this qualitative comparison.

Finally, from the fits in Table 3, the minimum track lengths $L_{10}$ required for 10% relative accuracy are determined. These are presented in Table 4.

Noteworthy findings include the following:

- Equation (8) due to Lenschow and Stankov (1986) and using the independently computed parameter values is remarkably accurate at predicting the random error for shorter track lengths.

- Empirical fits generally show a steeper decrease with track length than the $L^{-1/2}$ relationship predicted by (8), with exponents ranging from $-0.64$ to $-0.70$ at 10 m and 40 m, to nearly $-1$ for sensible and latent heat fluxes measured at a height 400 m. It is not clear whether the more rapid decay in error is primarily related to the finite domain size—with this issue being most likely problematic at 400 m—or to other departures from the idealized statistical model of turbulence assumed by Lenschow et al. (1994).

- For flights at and below 100 m, track lengths of approximately 30 km or less are sufficient to achieve 10% precision in sensible and latent heat flux estimates in this particular environment. Required track lengths for momentum flux $\tau$ are considerably longer; nearly 90 km at 100 m height.

- At a height of 400 m, no reasonable track length is long enough to adequately measure the low sensible heat and momentum fluxes encountered there.

### 6.1 Uncertainties in flux error determinations

The previous comparison with (8) assumed that the relevant integral lengths and correlations were the "true" values. In practice, it may be necessary when estimating sampling errors to determine these parameters empirically from the flight tracks themselves. These are thus subject to sampling errors of their own that in turn imply greater uncertainty in the estimation of $\sigma$.

To examine this problem, integral lengths $I_f$ and correlations $\rho$ were estimated directly from the flight tracks. Based on (8), an error bias factor was defined as

$$\Phi \equiv \left[ \frac{(\rho_{\text{est}}^{-2} + 1) I_{f,\text{est}}}{(\rho_{\text{true}}^{-2} + 1) I_{f,\text{true}}} \right]^{1/2}, \tag{11}$$

where the "est" subscripts refer to the track-estimated quantities and "true" refers to the domain-averaged quantities. Thus, the apparent sampling error is given as

$$\hat{\sigma} = \Phi \sigma, \tag{12}$$

where $\sigma$ is the "correct" flux error based on perfect knowledge of the integral length scale and correlation. Results are depicted for LH in Fig. 12. The solid lines depict the average bias factor as a function of track length; the dashed lines depict the standard deviation of the bias about this average.

Interestingly, at the lowest levels, there is an average low bias of between 10% and 20%, and this bias persists for the longest track lengths. This is likely an artifact of the method used, as the track determinations of integral length and correlation reflect the actual orientation of the track relative to the wind, whereas the generic "true" values do not. At higher altitudes, the directional dependence of $I_f$ disappears, as seen in Fig. 7g, and so does the apparent bias in the flux error.

At the lowest altitude of 10 m (Fig. 12a), the scatter about the mean bias quickly becomes small with increasing track length. Surprisingly, this is not the case at 100 m or 400 m. Rather, the relative uncertainty in the flux error is nearly constant with path length. Why this is the case requires further investigation.

Notwithstanding the non-negligible role of sampling uncertainties in $I_f$ and $\rho$ in estimating the flux uncertainties from (8), it is perhaps reassuring that the resulting variations are typically no more than about 10%, though larger errors are manifestly possible.

Similar results were found at lower levels for sensible heat and momentum flux (not shown). At 400 m, the scatter in the bias term for these variables is quite large, reflecting the small cross-correlations found near this level (see Fig. 8). When $\rho$ is small, even modest sampling errors in the determination of $\rho$ can produce large fluctuations in $\Phi$. In short, when sampling error is poor relative to the actual magnitude of the flux, even the ability to accurately characterize that sampling error is impaired.

## 6.2 The effect of track orientation

As previously noted, the algorithm for track definition, which in turn is based on the requirement for periodic (cyclic) tracks, does not permit the choice of track orientation relative to the wind direction. For this reason, it is not generally possible to use the methods described herein to empirically determine sampling error as a function of track length for tracks nearly parallel or perpendicular to the wind direction.

Having verified that (8) does appear to give reasonable results overall for the sampling error (though possibly overestimating that error for longer track lengths), it follows that the ratio $I_f/L$ largely determines the sampling error, all other factors being equal. From the directional dependence of $I_f$ alone, one may infer that flying crosswind should normally yield smaller errors for a given track length than flying parallel to the wind.

This assumption may be directly tested in one particular case. For the 7.2 km tracks, corresponding to $N = 1$, the tracks have an orientation of $\pm 45°$ and thus cross the mean wind direction of $307°$ at either $8°$ or $82°$. Results for these orientations should be practically indistinguishable from true parallel and crosswind flights.

Empirical flux errors were determined separately for these two cases, and the results are given in Table 5. With the exception of sensible heat at 400 m, where sampling error is in any case unusably large relative to the true flux, all flux variables are sampled with significantly greater precision in the crosswind direction. The difference is up to about a factor-of-two for latent heat up to at least 100 m and for all variables at the lowest level.

## 7 Conclusions

In this paper, the high-resolution large-eddy simulation of the marine boundary layer by Matheou (2018) was utilized to determine random flux sampling errors by a virtual aircraft flying tracks of various lengths at heights of 10 m, 40 m, 100 m, and 400 m. We eliminated the need for detrending by constructing these tracks to be cyclic with periods of the specified lengths.

The empirical results were compared with the theoretically derived expression of Lenschow and Stankov (1986). In support of these comparisons, we computed the required integral length scales $I_f$ and found that these are well described by expressions of the form $I_f = Az^b$ for $z \leq 100$ m, with coefficents $A$ and $b$ given in Table 2. The empirical exponents $b$ are generally between about 2/3 and 1 for fluxes of sensible heat, latent heat, and momentum, in contrast to Lenschow and Stankov (1986), who found exponents of 1/3 to 1/2. The reason for this apparent discrepancy is unknown. The commonly cited approximation for the integral length scale for fluxes given by (9) or (10) was also evaluated and found to consistently overestimate the true value by at least a factor of 2–5.

Using directly computed parameter values, (8) was shown to be remarkably accurate at predicting random errors for track lengths of approximately 10 km. However, it predicts a decay in error proportional to $L^{-1/2}$, whereas our empirical results show substantially more rapid decay in many cases. It is uncertain to what degree the difference is due to the finite domain size of the LES, to departures from the idealized statistical assumptions of Lenschow and Stankov (1986), or perhaps to a combination of both. It seems less likely that the first of these issues would affect the results for measurements at lower heights, as there is then very little contribution to total fluxes by wavelengths greater than 1 km, according to Fig. 6.

In general, track lengths of $\sim$30 km or less are sufficient for measuring sensible and latent heat fluxes in this simulated environment to better than 10% precision, provided that flights are conducted at or below 100 m. Substantially longer flight tracks ($\sim$85 km) are required for momentum flux, a result traceable to the low correlation $\rho_{U,w}$ (and thus low corresponding fluxes) between the horizontal and vertical wind components even near the surface. At 400 m height, only latent heat flux could be estimated with adequate precision using a flight leg of 35 km. On the other hand, legs in excess of 300 km would theoretically be required to estimate sensible heat flux or momentum flux with 10% precision.

For comparison, Sühring et al. (2019) found that 200 km flights were desirable to ensure 10% precision in sensible heat flux estimates within the lower half of the boundary layer. Their results were obtained for a deep, dry convective environment. The difference in specific findings undoubtedly reflects the larger integral length scales expected in that environment compared to the shallow, weakly forced marine boundary layer considered herein. It also highlights the difficulty of extrapolating findings from one environment to another.

Finally, it was empirically confirmed that for a relatively short, fixed track-length of 7.2 km, flux sampling errors are reduced up to a factor of two by flying crosswind rather than parallel to the wind, especially at lower levels, consistent with the typically shorter integral length scales associated with the crosswind direction.

*Code and data availability.* Data sets and Jupyter notebook files utilized in the analysis may be requested by sending a 1 TB USB drive to the corresponding author.

*Author contributions.* The author is solely responsible for the analysis and discussion in this paper.

*Competing interests.* The author declares that he has no competing interests.

*Acknowledgements.* George Matheou kindly provided the LES data set, without which this study would not have been possible. It was in turn created with support from the NASA High-End Computing (HEC) Program through the NASA Advanced Supercomputing (NAS) Division at Ames Research Center and by the University of Connecticut High Performance Computing (HPC) facility. The paper was significantly improved by comments and suggestions by Ankur Desai and by two anonymous reviewers. Partial support was provided by the Chequamegon Heterogeneous Ecosystem Energy-balance Study Enabled by a High-density Extensive Array of Detectors (CHEESEHEAD) project, NSF Award #1822420.

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

**Table 1.** Studies examining the sampling error problem in aircraft flux measurements with the aid of large eddy simulations (LES).

| Study | Model domain | Context | Applicable result or finding |
| --- | --- | --- | --- |
| Schröter et al. (2000) | $401 \times 401 \times 42$, $\Delta x = 50$ m, 2 km extent | Dry convective boundary layer | "flight duration of approximately 2000 s [at 100 m sec$^{-1}$] was necessary to obtain accuracy of 10% for the statistical error of the sensible heat flux in the lower half of the CBL..." |
| Sühring and Raasch (2013) | $1400 \times 1400 \times 100$, $\Delta x = 40$ m, 56 km extent | Heterogeneous land surface | included an examination of flux estimate errors from virtual flight legs through the LES domain |
| Sühring et al. (2019) | $1714 \times 2286 \times 500$, $\Delta x = 7$ m, 12 km $\times$ 16 km extent | Flux disaggregation over heterogeneous land surface | virtual flight segments of 200 m to 12.5 km at 49 m height |
| **This study** | $4096 \times 4096 \times 1200$, $\Delta x = 1.25$ m, 5.12 km extent | Cloud-topped marine boundary layer over homogeneous ocean surface | "instantaneous" (single time-step) virtual flight legs from 7.2 km to 102.5 km at heights of 10, 40, 100, and 400 m |

**Table 2.** Coefficients for the integral length scale $I_f \approx Az^b$, valid up to at least 100 m altitude in this simulation, depending on the variable (see Fig. 7), where both $I_L$ and $z$ are in meters. Also given is the approximation $I_f \approx \bar{A}$ for selected variables.

| | | $A$ | $b$ | $\bar{A}$ |
|---|---|---|---|---|
| $q$ | Crosswind: | 20.6 | 0.51 | |
| | Parallel: | 35.2 | 0.41 | |
| $T$ | Crosswind: | 20.5 | 0.56 | |
| | Parallel: | 33.1 | 0.36 | |
| $U_{\parallel}$ | Crosswind: | 309 | $-0.06$ | 278 |
| | Parallel: | 146 | 0.09 | 206 |
| $U_{\perp}$ | Crosswind: | 161 | 0.11 | 233 |
| | Parallel: | 387 | $-0.03$ | 371 |
| $w$ | Crosswind: | 0.63 | 1.03 | |
| | Parallel: | 2.95 | 0.82 | |
| SH | Crosswind: | 0.53 | 0.97 | |
| | Parallel: | 3.01 | 0.64 | |
| LH | Crosswind: | 0.50 | 1.00 | |
| | Parallel: | 2.85 | 0.69 | |
| $\tau_{\parallel}$ | Crosswind: | 0.91 | 0.81 | |
| | Parallel: | 2.88 | 0.70 | |
| $\tau_{\perp}$ | Crosswind: | 0.86 | 0.85 | |
| | Parallel: | 2.34 | 0.79 | |
| *TKE* | Crosswind: | 77.3 | 0.06 | 96 |
| | Parallel: | 95.4 | 0.05 | 115 |

**Table 3.** Coefficients for the empirical fits depicted as dashed black lines in Fig. 11, where the flux uncertainty $\sigma_f = BL^c$, and $L$ is the track length in kilometers. Resulting units are W m$^{-2}$ for latent and sensible heat fluxes; N m$^{-2}$ for horizontal momentum flux.

| Height | Sensible Heat | | Latent Heat | | Horiz.Momentum | |
|---|---|---|---|---|---|---|
| | $B$ | $c$ | $B$ | $c$ | $B$ | $c$ |
| 10 m | 10.2 | $-0.64$ | 74.2 | $-0.65$ | 0.052 | $-0.65$ |
| 40 m | 15.5 | $-0.64$ | 136.6 | $-0.69$ | 0.135 | $-0.70$ |
| 100 m | 26.9 | $-0.75$ | 257.5 | $-0.79$ | 0.247 | $-0.78$ |
| 400 m | 44.8 | $-0.95$ | 525.1 | $-0.98$ | 0.411 | $-0.80$ |

**Table 4.** Minimum track length $L_{10}$ in kilometers required to achieve 10% uncertainty in flux estimates based on the expression and coefficients in Table 3.

| Height | Sensible Heat | Latent Heat | Horiz. Momentum |
|---|---|---|---|
| 10 m | 9.0 | 9.1 | 18.0 |
| 40 m | 19.1 | 19.3 | 60.5 |
| 100 m | 33.1 | 29.7 | 85.7 |
| 400 m | 334.9 | 34.5 | 410.9 |

**Table 5.** For 7.2 km crosswind and parallel tracks, the mean error (bias) and standard deviation $\sigma$ in simulated flux measurements.

| Height | | Sensible Heat [W m$^{-2}$] | | | Latent Heat [W m$^{-2}$] | | | Horiz. Mom. [N m$^{-2}$ 10$^{-2}$] | | |
|---|---|---|---|---|---|---|---|---|---|---|
| | | True | Mean Error | $\sigma$ | True | Mean Error | $\sigma$ | True | Mean Error | $\sigma$ |
| 10 m | Crosswind | 24.7 | −0.17 | 2.02 | 176.0 | −1.11 | 14.79 | 7.98 | 0.125 | 1.04 |
| | Parallel | | −0.49 | 3.88 | | −3.64 | 27.82 | | 0.079 | 2.05 |
| | Ratio | | | 1.92 | | | 1.88 | | | 1.97 |
| 40 m | Crosswind | 23.3 | −0.25 | 2.84 | 177.7 | −1.85 | 22.91 | 7.55 | 0.437 | 2.76 |
| | Parallel | | −0.68 | 4.32 | | −6.03 | 38.89 | | 0.898 | 3.94 |
| | Ratio | | | 1.52 | | | 2.05 | | | 1.43 |
| 100 m | Crosswind | 19.8 | −1.00 | 3.97 | 175.4 | −8.68 | 36.18 | 7.69 | 1.18 | 3.94 |
| | Parallel | | −0.89 | 4.77 | | −9.12 | 46.29 | | 2.51 | 5.46 |
| | Ratio | | | 1.20 | | | 2.13 | | | 1.39 |
| 400 m | Crosswind | 1.8 | −0.44 | 5.60 | 161.8 | −6.24 | 48.63 | 3.31 | 4.05 | 3.90 |
| | Parallel | | −0.04 | 4.08 | | −17.52 | 57.79 | | 6.78 | 4.32 |
| | Ratio | | | 0.73 | | | 1.67 | | | 1.11 |

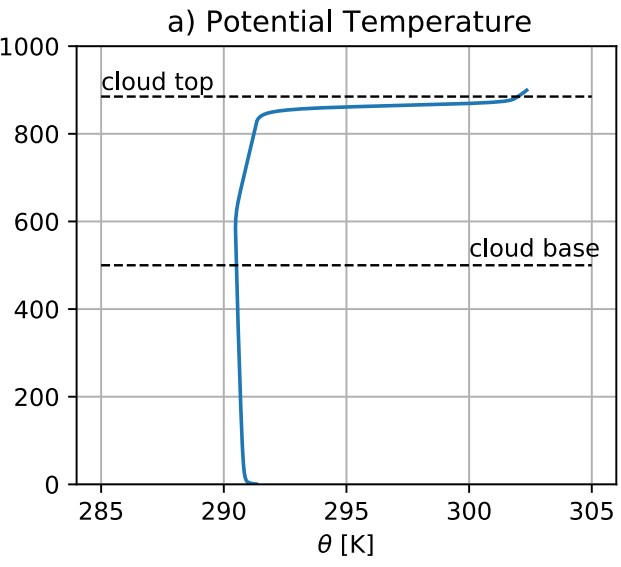

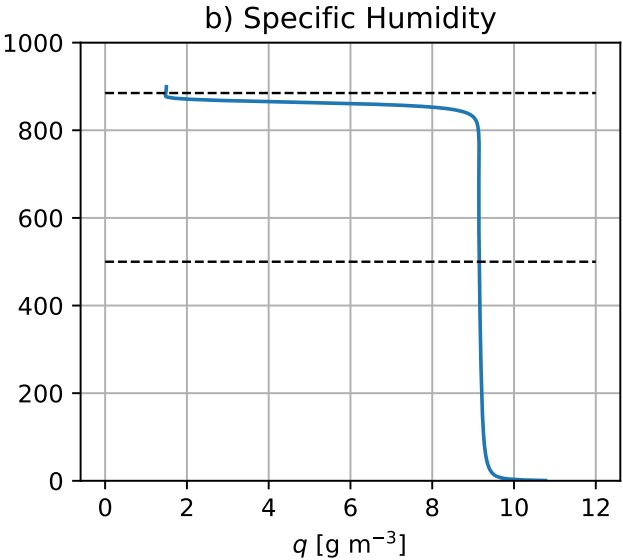

**Figure 1.** Domain-averaged profiles of a) potential temperature $\theta$ and b) specific humidity $q$ at the end of the LES run. Dashed lines indicate the heights of the minimum and maximum altitudes at which non-zero cloud water occurred at any point in the domain.

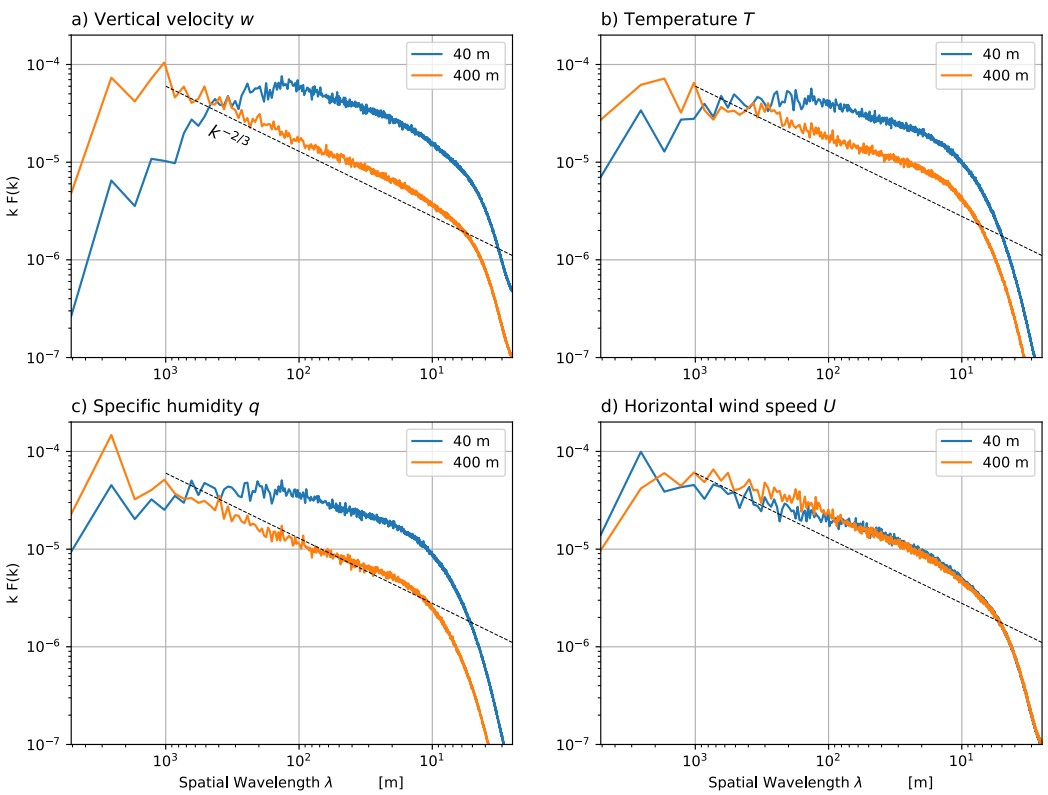

**Figure 2.** Radially averaged power spectra for selected scalar variables at two representative heights 40 m = $0.05z_i$ and 400 m = $0.48z_i$: a) vertical velocity, b) temperature, c) specific humidity, d) horizontal wind speed. For reference, black dashed lines depict the power-law exponent of $-2/3$ expected for a Kolmogorov spectrum in the inertial subrange.

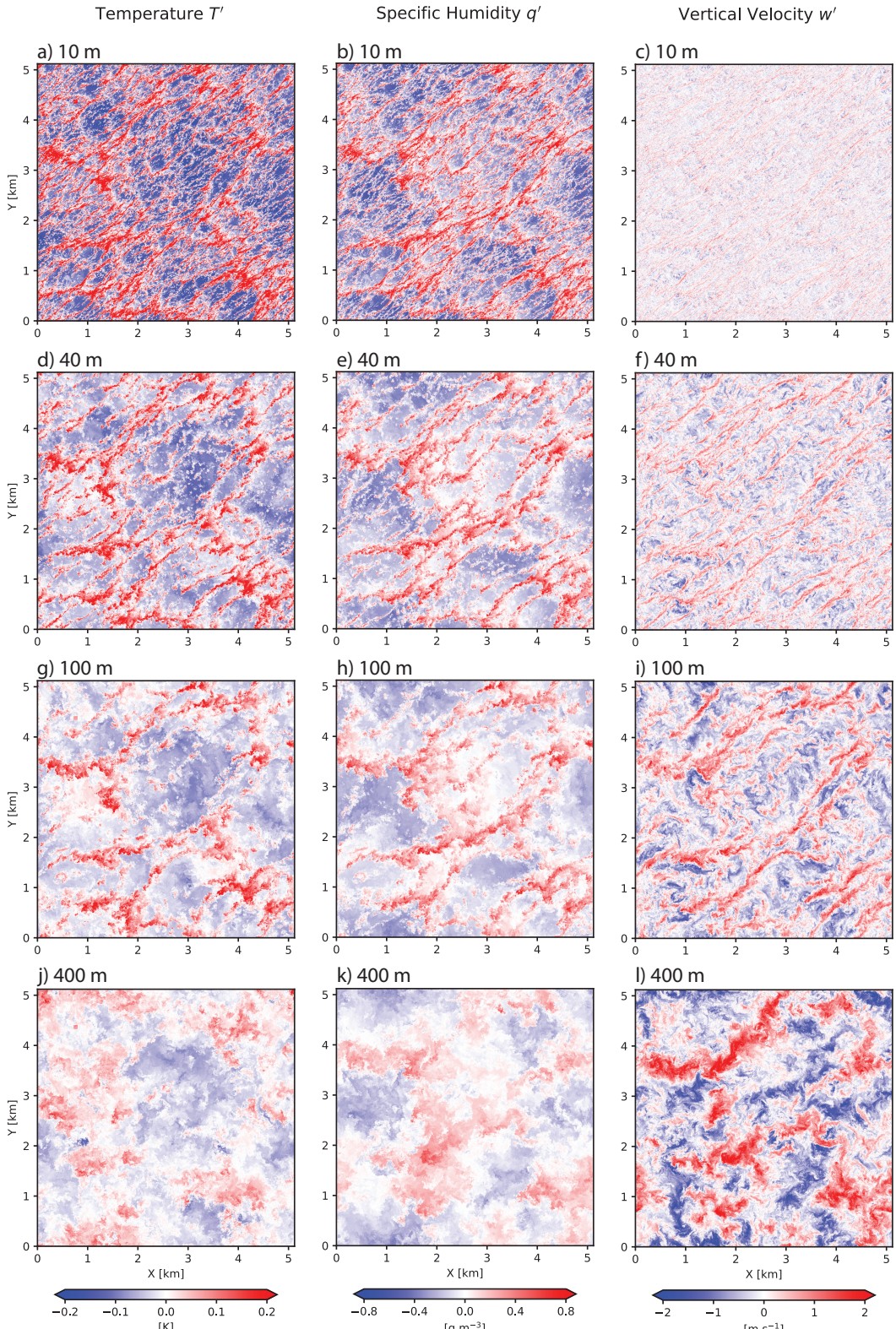

**Figure 3.** Horizontal cross-sections of $T'$ (left column), $q'$ (center column), and $w'$ (right column). Rows correspond to 10 m, 40 m, 100 m, and 400 m heights.

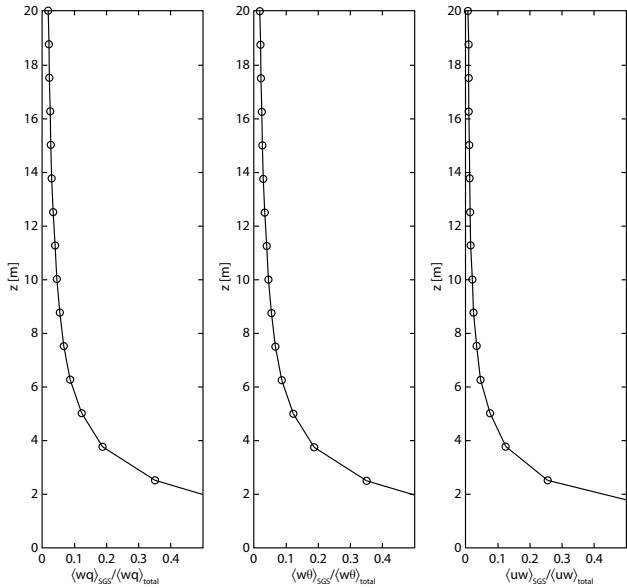

**Figure 4.** Parameterized subgrid-scale flux as a fraction of total turbulent flux for the LES used in this study. Left: latent heat. Center: sensible heat. Right: horizontal momentum (figure courtesy G. Matheou, pers. comm.)

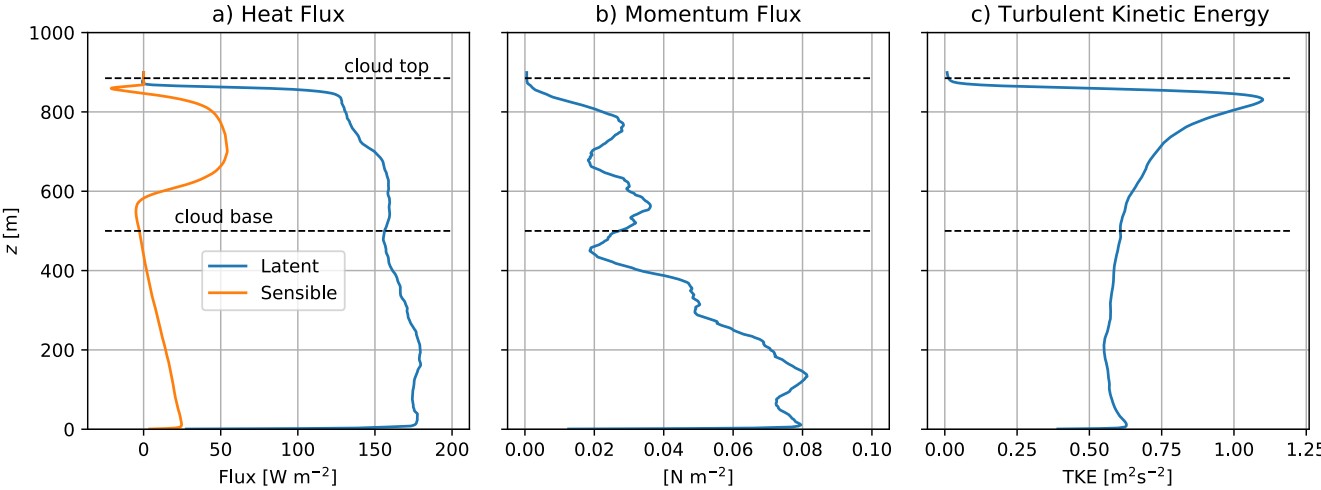

**Figure 5.** Domain-averaged turbulent quantities computed from grid-resolved $u'$, $v'$, $w'$, $T'$, and $q'$ in the LES simulation.

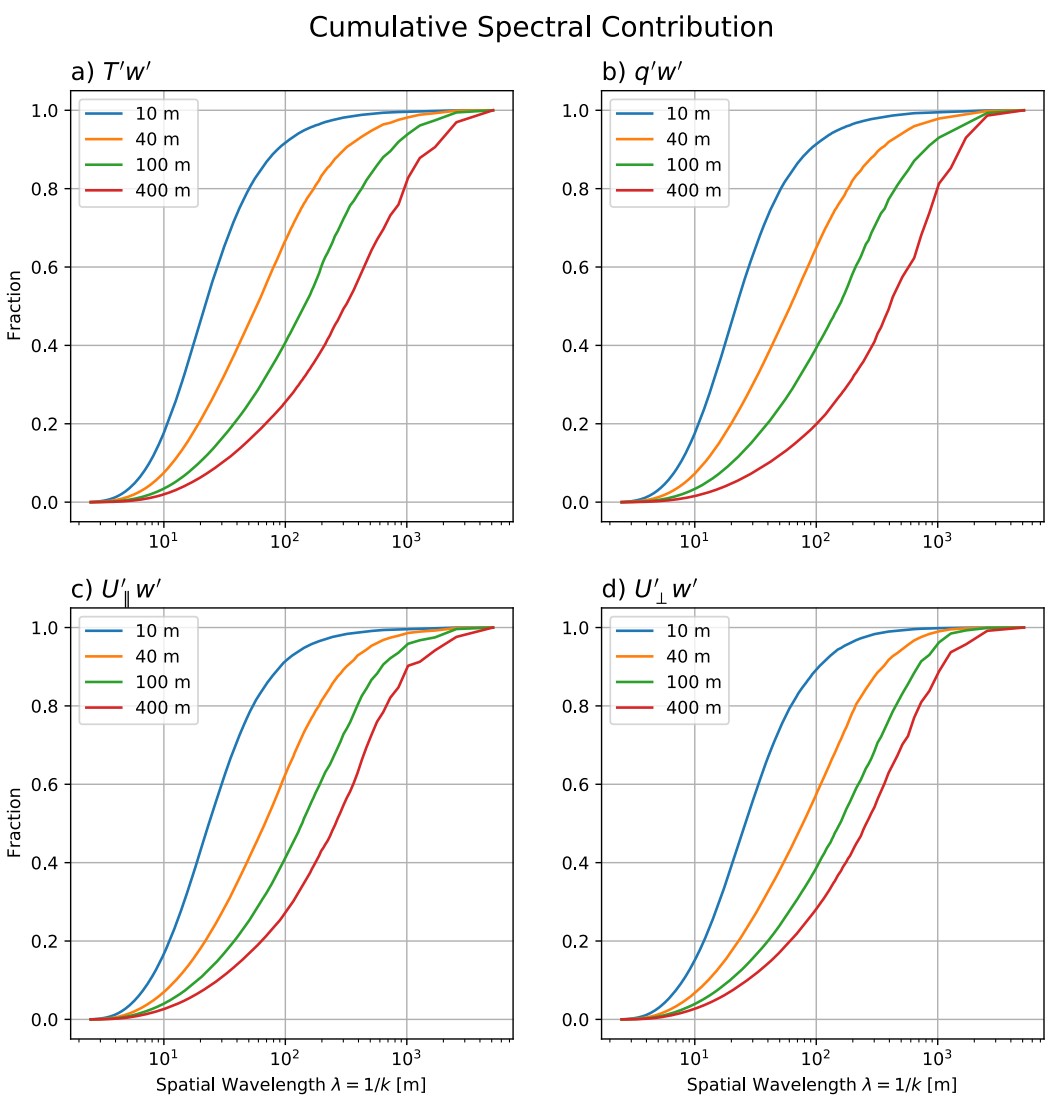

**Figure 6.** Ogive plots of the spectral contributions to the total eddy correlation fluxes for a) sensible heat, b) latent heat, c) horizontal momentum in the along-wind direction, d) horizontal momentum in the crosswind direction.

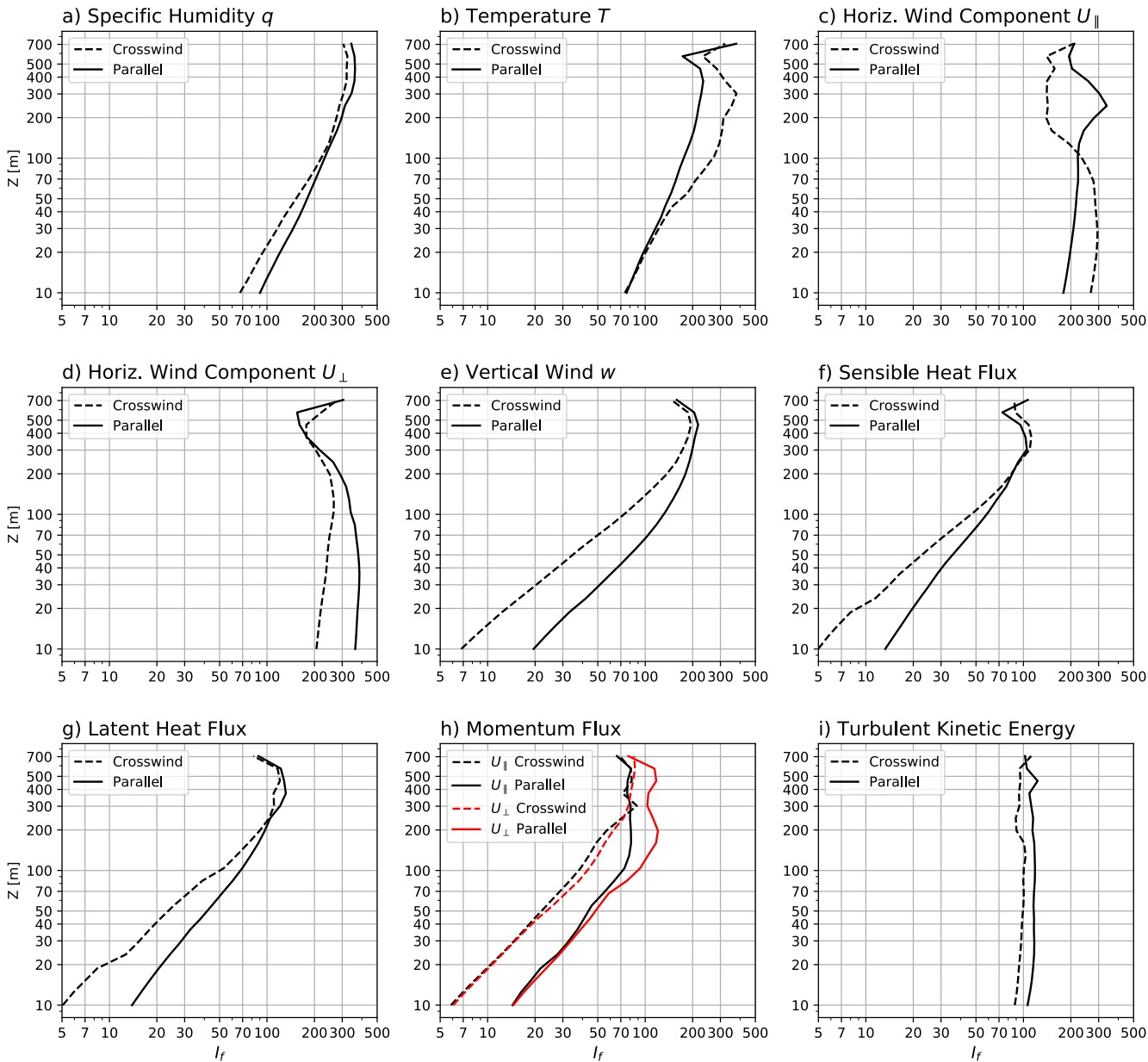

**Figure 7.** Integral length scales $I_f$ for the indicated quantities, measured along ("parallel") and perpendicular to ("crosswind") the mean wind direction.

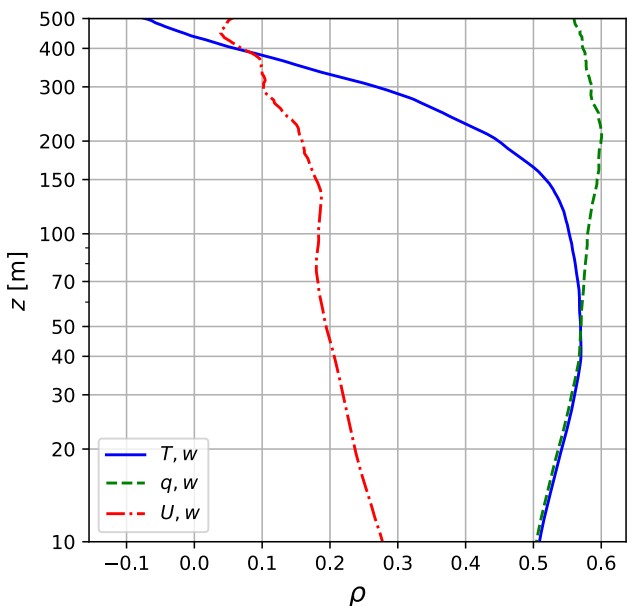

**Figure 8.** Sub-cloud profiles of correlations between vertical velocity $w$ and temperature $T$, specific humidity $q$, and horizontal wind speed $U$.

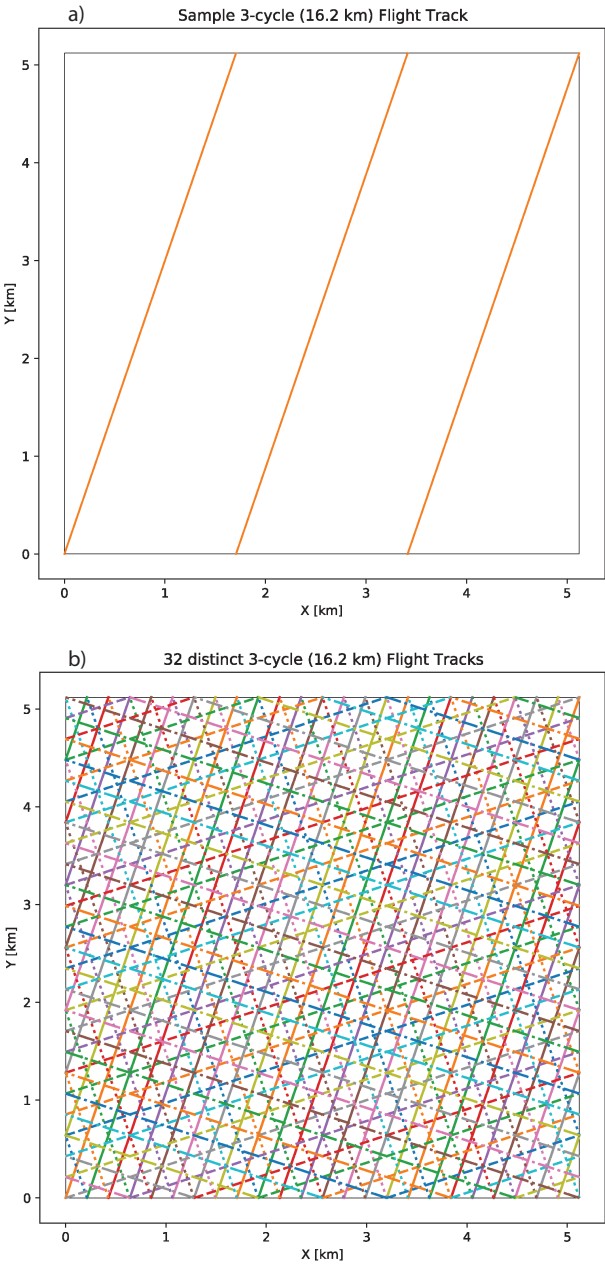

**Figure 9.** Example of flight tracks constructed to pass through the LES domain an integral number of cycles before exactly repeating. a) A single 3-cycle (16.2 km) track. b) All possible realizations of 3-cycle tracks such that a minimum parallel spacing of 200 m between track legs is maintained.

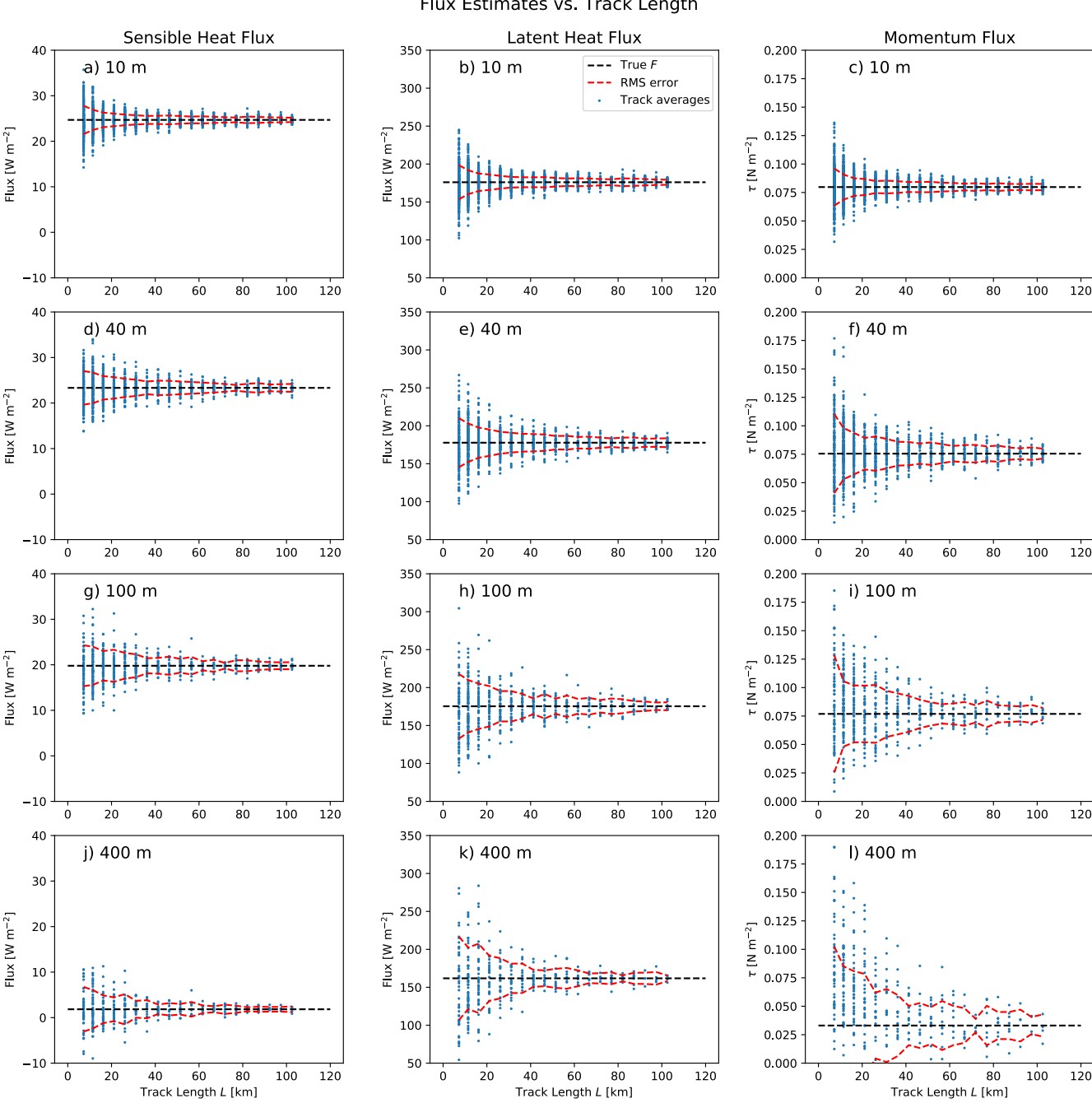

**Figure 10.** Flux estimates vs. track lengths for sensible heat (left column), latent heat (center column), and horizontal momentum (right column). Red dashed curves depict the root-mean-squared deviation of estimates from the true domain-averaged flux (horizontal dashed line).

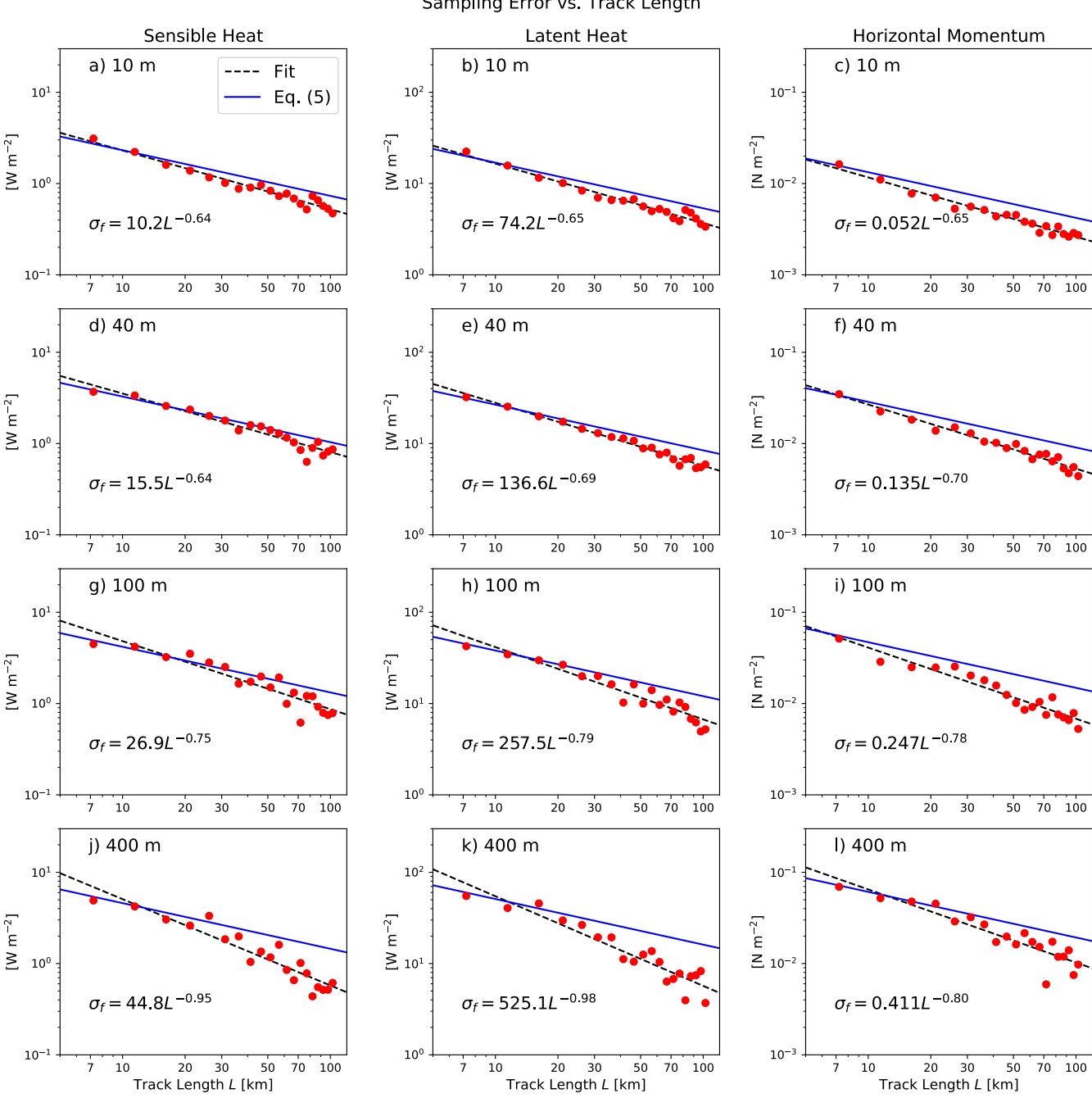

**Figure 11.** Relationship between ensemble RMS error and track length for sensible heat flux (left column), latent heat flux (center column), and momentum flux (right column). Red dots depict root-mean-squared deviation of estimates from the true domain-averaged flux, corresponding to the red curves in Fig.10. An empirical power-law fit is depicted by the dashed black line, with the coefficients of the fit indicated in the expression for $\sigma_f$. The solid blue lines represent (8) using the parameter values given in Table 3.

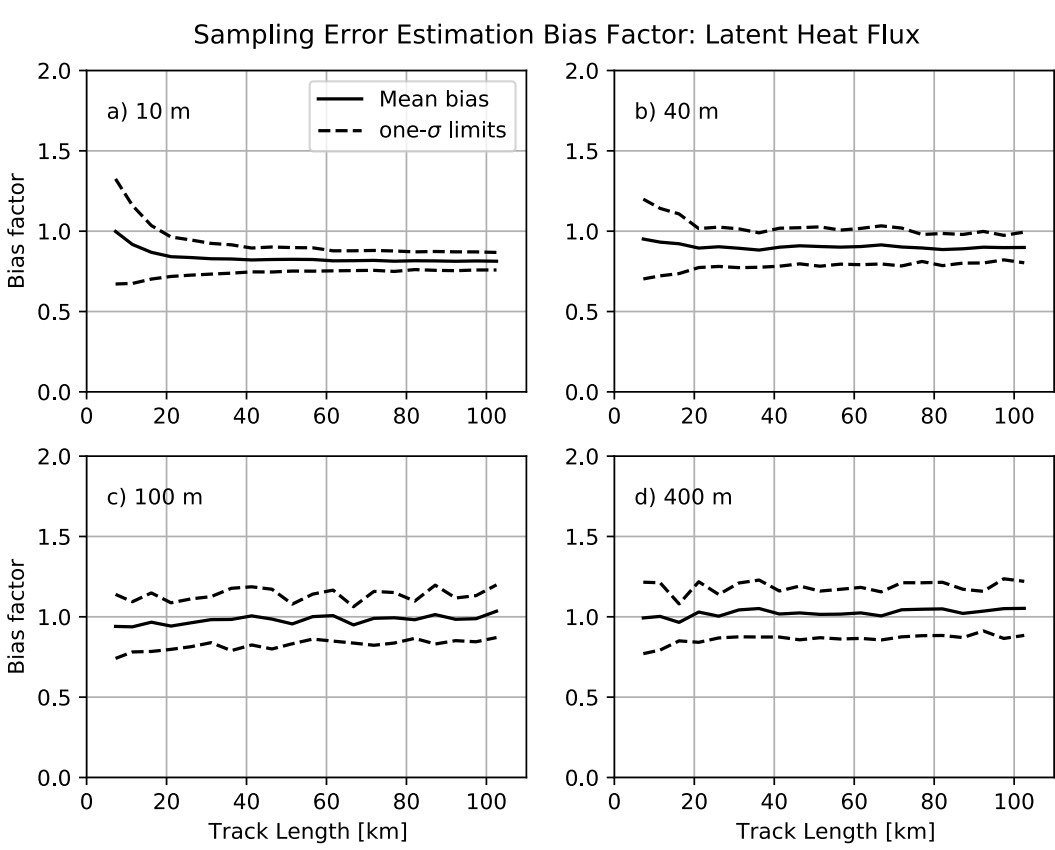

**Figure 12.** Distributions of empirically determinated bias factors from (11) as functions of track length.