# Peer review of "Sampling Error in Aircraft Flux Measurements Based on a High-Resolution Large Eddy Simulation of the Marine Boundary Layer"

_Atmospheric Measurement Techniques, 2020_

## Referee Comment (RC1) · Anonymous Referee #1 · 22 Sep 2020

**Overview**

In this study, the author uses an output from a Large-Eddy Simulation performed with a high-resolution, which explicitly resolved the major part of the turbulent structures. The domain-averaged turbulent fluxes can thus be considered as the "truth". This reference is then used to explore, with virtual track traversing the LES domain, the capacity of aircraft flux measurements to properly estimate the turbulent fluxes, the associated sampling problem and random errors. The results, compared with previous work on the topic (Lenschow and Stankov (1986), Lenschow et al., (1994)), lead to similar conclusions about the required length of the tracks to limit the sampling errors.

**General comments**

- Regarding the scope of AMT scientific questions, the question of the publication of this study in this journal may be raised. Indeed, even if the topic is about airborne flux measurements, the study is based exclusively on results from numerical simulations. It is regrettable that no observations are used in this study, either to be confronted with the simulation output or to apply the results obtained, for example on past measurement campaigns.

- The track definition used by the author can lead to flight tracks greater than the domain size thank to the cyclic boundary conditions of the LES. Nevertheless, as mentioned by the author, the finite LES domain is not able to reproduce structures greater than the domain size. Is a domain of 5,12 x 5.12 km$^2$ is therefore large enough to study airborne sampling and eddy correlation flux estimation? With a larger domain size, the characteristics of the simulated turbulent structures may be different. With this issue of the limited size of the domain, does a LES with a larger mesh grid and a larger domain have been appropriate? Taking the example of the University of Wyoming KingAir aircraft mentioned by the author, with a true air speed of 85 m/s and a measurement frequency of 25 Hz, the sampling spatial resolution is then about 3.5m. Thus, a grid mesh 3 times larger than the one used here could be adequate.

- There are many figures (19 in total), some of which seem redundant or could be concatenated. Several of them are simply mentioned in the text without being analyzed or discussed. The question of the relevance of these figures may arise, not helping to clarify the main message of the article. It obviously seems appropriate and necessary to present the simulation with the help of a few figures, however, it is only from figure n°13 that the central purpose of the paper begins to be addressed.

**Specific comments**

**Introduction:**
- The works of Lenschow and Stankov (1986), Lenschow et al. (1994), and Mann and Lenschow (1994) were not only based on theoretical considerations and statistical models but also on observations. It might be useful to include in the introduction, some studies on experimental data and field campaign. In general, the introduction could be enhanced in terms of bibliographic references, such as Brooks and Rogers (1997) Cook and Renfrew (2015) or Brilouet et al. (2017).

- Line 34: The LES is able to resolve explicitly the major part of the turbulence but it remains a sub-grid contribution. Even if with a 1.25m resolution, this contribution becomes rapidly negligible with the altitude, it might be useful to mention that total turbulence = explicitly resolved + subgrid contribution. - After line 42, it is not clear if we are still in the introduction section or if the section "description of the method" has already started. It would be useful if the main goal of the study could be more clearly highlighted and if an outline of the article were provided at the end of the introduction before going into the details of the simulation and the method.

- Line 60: It is correct that using a LES to examine the aircraft flux sampling problem in MABL is unique. Nevertheless, it can be mentioned that previous studies compared LES outputs with airborne measurements such as Brilouet et al. (2020) even if the resolution was coarser.

**Data:**
- The case study is from the field campaign DYCOMS-II, Are there any observations that might be relevant to the study?

- The case study is a nocturnal cloud-topped marine boundary layer. When the author describes the environment, a few elements describing the main characteristics of this type of stratocumulus condition could be instructive for the reader (such as $z_i$ at the cloud top, the strong inversion with entrainment at the cloud top, . . .).

- Figure 2: the figure is rather small. The units of the power spectra are not mentioned. Does it might be interesting to present normalized spectra (by the variance:

$kF(k)/\sigma_X^2$)? Does the spatial wavelength is $\lambda = 1/k$ or $\lambda = 2\pi/k$?

- Line 95: It might be interesting to compare with previous works.

- Lines 96-98: At 40m height (0.05 zi), this is the surface layer. How much the turbulence is explicitly resolved at this height? What is the vertical profile of TKE resolved / total TKE? Also, the surface layer may have different characteristics than the layer above. Does the Monin-Obhukhov Similarity theory (MOST) is available? It would be interesting to enhance the discussion with some references on the turbulent structure inside the surface layer such as Katul et al. (2011) or Sun et al. (2016).

- Lines 99-100: Do the spectra of temperature and specific humidity reveal more energy at longer wavelength due to the influence of mesoscale on those parameters? If the domain was larger, would the wavelengths be longer?

- Line 101: What is the reason that the horizontal wind speed spectrum has no significant dependence on height?

- Lines 104-105: The author has chosen four representative heights, one at 10 m and another at 40 m. Are these heights characteristic of airborne measurements?

- Figures 3-5: 3 figures are considered for 4 lines. It would be interesting to concatenate them into a single figure. It will be easier to compare the characteristics of each parameters and their evolution with the height (for example with left panels at 10 m, middle panels at 40 m and 100 m and right panels at 400 m with a parameter by row).

- Lines 106-110: Also, a link with previous work would be valuable.

- Figure 6: Is this figure really essential to the article?

- Line 114: It might be helpful to define the sensible (H) and latent (E) heat fluxes. Commonly, the E notation refers to the surface moisture flux or evaporation ($E = \rho \times w'q'$). Perhaps the LE or LvE notation is more appropriate for the latent heat flux.

- Lines 114-119: Is the definition of sensible and latent heat fluxes and their expressions as a function of fluctuations valid at different altitudes in the boundary layer? Is it not defined only for surface exchanges? The sensible heat flux is the amount of heat exchanged between the surface and the atmosphere and the latent heat flux represents the energy released or absorbed during a phase change. I may be mistaken and in that case, I apologize for this unwelcome comment.

- Figures 7 and 8: These figures are not described or analyzed in the article. Are they essential to the article?

- Line 130: It would be interesting to explain the TKE profile and how this is expected, in terms of the processes involved, given the case study under consideration. Here again, a connection with previous studies on this subject would be appreciated.

**Integral length scales**
In this section, the work of Lumley and Ponofsky (1964) could enhance the bibliography as a pioneer on these issues.

- Line 140: It is the first time, since the introduction, that the random error is mentioned. As this is the main focus of the article, wouldn't it be a good idea to highlight it further? The current design of the article suggests that it is secondary to the integral scales.

- Line 149-150: To introduce the random error in a simplified point of view, is the equation 1 of Lenschow and Stankov can be relevant?

- The spatial correlation $\rho_{w\psi}$ is defined twice (line 156 and line 164).

- Line 158: In order to specify the experimental difficulties in estimating the integral length scale, the study of Durand et al. (2000) could be instructive.

- Figure 12: Even if the random error definition contains the correlation $\rho_{w\psi}$, is the figure really essential to the article?

**Simulated aircraft measurements**
- Lines 208-209: This sentence perfectly summarizes the main topic of the study. Isn't

it a bit lateÂă? This message does not appear clearly enough throughout the article.

- Lines 246-247: As mentioned in the general comments, I have some concerns about the domain size with respect to the characteristic scales of fluxes that can be observed during airborne measurement campaigns. Consequently, the results that will arise from this study seem difficult to be transposed to measurement campaigns.

- Line 249: Another way to check Taylor's hypothesis, for airborne measurements, the true air speed (here $V = 85m/s$) can also be compared to the intensity of the turbulence $(\overline{u'^2})^{1/2}$. If $V \gg (\overline{u'^2})^{1/2}$ then the statistical properties of the turbulence field are assumed to be unchanged over the considered time interval.

**Results**
- Figures 14-16: These three figures could be concatenated into one. Moreover, even if these figures are at the core of the study, they are barely detailed and analyzed (Figure 15 is barely mentioned).

- Line 261: Including bibliographic references would be valuable.

- Figures 17-19: In order to facilitate the understanding of the figures, it can be useful to keep the empirical RMS error in red rather than changing the color. Are the parameters in blue necessary? If so, would it be better to include them in a table? As the minimum track length L10 for $10\%$ relative accuracy is one of the main results, would it be a useful to group them together, for each flux and each altitude, in a table?

References:
Brilouet, P.-E., Durand, P., and Canut, G. (2017), The marine atmospheric boundary layer under strong wind conditions: Organized turbulence structure and flux estimates by airborne measurements, J. Geophys. Res. Atmos., 122, 2115– 2130.

Brilouet, P.-E., Durand, P., Canut, G. et al (2020). Organized Turbulence in a Cold-Air Outbreak: Evaluating a Large-Eddy Simulation with Respect to Airborne Measurements. Boundary-Layer Meteorol 175, 57–91.

[Figure]

Brooks, I. M., and D. P. Rogers. (1997). Aircraft Observations of Boundary Layer Rolls off the Coast of California. J. Atmos. Sci., 54, 1834–1849.

Cook, Peter A., and Ian A. Renfrew. (2015). Aircraft‐based observations of air–sea turbulent fluxes around the British Isles. Quarterly Journal of the Royal Meteorological Society 141, no. 686 139-152.

Durand, P., Thoumieux, F. and Lambert, D. (2000). Turbulent length‐scales in the marine atmospheric mixed layer. Q.J.R. Meteorol. Soc., 126: 1889-1912.

Katul, Gabriel G., Konings, Alexandra G. and Porporato Amilcare (2011). Mean Velocity Profile in a Shared and Thermally Stratified Atmospheric Boundary Layer. Phys. Rev. Lett. 107, 268502

Lumley J. L. and H. A. Panofsky. (1964), The structure of atmospheric turbulence. New York (Interscience Publishers), 239 pp.

Sun, J., and J. R. French, (2016): Air–Sea Interactions in Light of New Understanding of Air–Land Interactions. J. Atmos. Sci., 73, 3931–3949
* * *

---

## Referee Comment (RC2) · Anonymous Referee #2 · 19 Oct 2020

Based on data from a large-eddy simulation for a stratocumulus topped marine boundary layer, the author performed an ensemble of flight measurements and analyzed the convergence of the sampled fluxes towards their truth, as well as investigated the dependence of the random error on the track length. The author compares the observed random error against the theoretically derived expression by Lenschow and Stankov (1986) and found good agreement for track length of 10-30 km. Further, integral length scales of the turbulent quantities were calculated from the LES data for different flight track angles. The author shows that integral length scales depend on the flight track angle and compares these with the proposed, and still commonly-used, approaches by Mann and Lenschow (1994).

The topic and the content of the paper fits well into the journal and is of high interest to the research community. Due to lack of any alternative ways to estimate the random error for flight measurements, almost everybody uses the expressions proposed by Lenschow and Stankov (1986), though it is already known that especially for shorter tracks the estimated random error often does not reflect the true uncertainty. Here, especially for shorter flight tracks, an improved random error estimation is highly desired in order to avoid misinterpretations of observations.

The paper itself is well written and results are sufficiently presented, though at some points in the text more information needs to be given. At several points in the text it is not clear how variables are calculated and from which data, i.e. from the full 3D LES data or from the sampled space-series along the flight tracks. Also, at some points the discussion is rather short and the findings are not well put into the context of previous research. For example, also the study by Schröter et al. (2000) had already analyzed how the sampled flux converges towards its truth with increasing track length. Even though the atmospheric setups are not comparable, the findings presented here should be put into the context of previous work.

After extensive review I can recommend the manuscript for publication only after major revisions have been done and extended analysis is presented. My major concern is outlined in the following.

(A) I miss one important aspect in the study. The author shows that the predicted random errors according to the Lenschow and Stankov formula matches the observed standard deviation remarkably well. However, in this study the random error using the LS86 formula is calculated based on integral length scales, correlation coefficients, and fluxes, that were inferred from the three-dimensional LES data. Though it is nice to show that the LS86 formula works well in the theoretical case where all these data is available, this approach does not reflect the reality at all. In reality, the integral length scale need to be calculated based on the flight-sampled data itself, hence, it is exposed to the same sampling errors as the flux is. Particularly for short track lengths,

the errors in the integral length scales are supposed to be remarkably high. These biased values then propagate into the LS86 formula, with the consequence that the standard deviation of the predicted random error is also very high, with far reaching implications concerning the interpretation of the measurement, as the random error estimation cannot be trusted anymore for short tracks. In some situations the LS86 formula indicates high random error, but in other situations it indicates low random errors (please see the discussion in Sühring and Raasch (2013) about this). Hence, the comparison of the LS86-predicated error against the truly random error in the way you have done it here is actually not fair and does not help much in the interpretation of observed data. The manuscript would strongly profit if you add such an analysis. I would propose to add following analysis: * How does the integral length scale and the correlation coefficient used in Eq. 5 depend on the track length? Here a direct comparison against the true value is possible since the integral length scales from the 3D LES data are available. * How does the random error behaves when it is calculated directly from the sampled space-series, and how well does it match with the 'true' random error? Based on this, the performance of the LS86 formula can be directly shown for different track length, which would be extremely helpful for researchers.

Further comments:

Abstract - Line 2: vertical turbulent fluxes

Line 26 - 30: The paragraph is quite imprecise, it is not clear to what refinements done by Mahrt (1998) or findings by Hollinger and Richardson (2005) the author refer to. The introduction would profit if the author would elaborate this a bit more.

line 44: Please remove the word "exceptional". Though it is indeed quite a large setup, it is not exceptional any more. In the last one to two years such setups have become already standard in the LES community.

line 42-44: To the reader of the manuscript it might appear unclear what is meant by this sentence "The single most ...". I suppose the author mean that the smaller scales

in the LES are filtered by the subgrid scheme, as well as by numerical errors, being not sampled adequately in an LES.

line 48-49: I disagree with the last part of the sentence. Emulating turbulence measurements in an LES always suffers from the missing subgrid-scale contributions, numerical errors etc., independent which grid resolution is used. Comparing measured turbulence spectra and spectra derived from LES-sampled data will always show a drop-off spectral energy on the smallest spatial scales (about 10 times the grid spacing as a rule of thumb), as it is also the case here (Fig. 2). The relevant question is if, and when, how much the missing subgrid-scale contributions affect the analysis of sampled turbulence data. I would recommend just to rephrase the part with the subgrid-scale flux here.

end of introduction: I miss a manuscript outline here, to guide the reader through the manuscript.

line 79-88: Beyond the fact that a marine boundary layer was simulated, what is the general atmospheric setup? Is this a boundary layer in the trade-wind zones or in a polar region (which latitude). Of course, according to Fig. 1 it becomes clear, but such information should be also given in the text.

line 93: How were these spectra calculated? Were they calculated from the emulated flights or from the 3D LES data (1D or 2D spectra)?

Fig. 2: Is it temperature or potential temperature? In Fig. 1 profiles of potential temperature are shown, later on the author only refer to temperature, though I cannot find any statement about a transformation.

line 98: With the phrasing "circular eddies" you indirectly imply that the turbulence is isotropic, which isn't the case as plenty of observations and simulation data show. I would recommend to simply remove the word "circular" here.

line 99-103: The author describes the spectra sufficiently here, though some references with respect of the minimum domain size of the LES domain are missing. However, I miss some further discussion about possible implications on further results. It is well known that at smaller wavelength the spectral resolution is bounded due to the subgrid-scale model as well as numerical dispersion and dissipation errors, causing these steep drop-off. In most cases this in no big issue as the smaller scales do not contain much energy. But also at the longer wavelengths, the spectral resolultion of LES is bounded by the domain size, especially for humidity (de Roode et al. 2004). From previous studies it is known that structures grow in time, meaning that the spectral peaks move towards larger wavelengths, until the structures cannot grow anymore as they are bounded by the domain size. Somehow the spectra for q indicates this. In case this happens I suppose the integral length scales of humidity are also affected by this, though it won't change much on the overall results I guess. You already bring this up in line 135-137, but maybe it is worth to already bring this up here too.

line 91/104: Though it is only one author, the we-form is repeatedly used throughout the manuscript.

Fig. 3: Again, is this absolute or potential temperature?

line 109-11: I disagree with this. The alignment with the mean wind is also visible in q' and T'. w' and q' (or T') correlate actually fairly well as it is typical in a convective boundary layer where the updrafts are created by buoyancy (which in turn is related to q' and T').

Eq. 1-2 and Fig. 7-8: Here it would be good to already mention that this quantity will be used for the integral length scale calculation. Without this background, which is not clear at this point in the manuscript, this might puzzle some readers. I am not entirely sure what these quantities are actually represent. To my understanding it is the local vertical flux at one point in time which can only be calculated from simulation data in the special case of a horizontally homogeneous boundary layer. To obtain heat fluxes in the traditional sense, time-averaging would need to be applied on top. Hence,

w'T' is not a real flux but a quantity used to compute integral length scales. I would recommend to make this clear in the manuscript, i.e. that these temporal local fluxes cannot be compared one to one to flux measurements from aircrafts or towers, neither with respect to the spatial pattern nor with respect to the amplitude. But in the context of the integral length scale calculation such equations and plots does make sense.

line 128: one dot too much

line 128-129 and Fig. 9: As the author have already mentioned in the text the momentum-flux profiles look quite noisy which is due to the lack of time averaging. Especially for the momentum flux longer averaging periods are required to obtain smooth profiles. Does the author have access to time-averaged profiles in order to check whether the shown momentum flux profiles changes much? According to the wind profiles shown in Fig. 1, I suppose the momentum flux profiles would decrease linearly with height. Here it would be worth to note this in the text and maybe a statement of how much the comparison against emulated flight measurements will be affected, as the truth in momentum flux is just a first guess.

line 133-134: I am wondering why the lower range for the 90-% contribution changes among the different height levels. To calculate an ogive you actually start at the smallest wavelength (or largest wavenumber) and integrate until the target wavelength. The smallest wavelength, however, is fixed by the cut-off wavelength which depends on the grid spacing, so the intervals should be always something like [2.5m-x1], [2.5m-x2], etc.

Fig. 11 f-h: It is not entirely clear how the integral length scales for fluxes are obtained. I suppose you calculated $l\_f$ from the local fluxes obtained from equation 1-3, but it is nowhere written in the text. Or did you used equation 6-7 to calculate $l\_f$?

line 196-201: It would be nice to add the resulting length scales from equation 6-7 into Fig. 11 f-h to have this also visually.

Section 3.3 and 3.4: The information content is not that big to put it into separate sections, you can easily merge it with section 3.2. But that's of course a matter of taste.

line 211: Please add the interval for n directly after the arctan-formula, this will make it easier to understand what is done here.

line 208-215: Before you start with your flight-track setup, you should first start to explain this special problem which is just because you use a stationary frozen flow field rather than a moving framework with advection where this problem will not occur unless the domain size is large enough. This will make it easier for the reader to understand what is done here.

Fig. 13: Just a minor point, but it would be nice to mention the flight direction - from southwest to the northeast, or was it the other way around?

line 234-240: I am not entirely sure what is the effect of the insufficiently sampled true value due to lack of temporal averaging, especially for the momentum flux which has been clearly not converged. But as you compare the flight-sampled data against these true flux, and both are calculated based on the same data set, I suppose that this won't have any effect on the following results. But interested readers might ask the same question, so it could be worth to spend one or two sentences here to make this more clear.

244-245: It is unclear which bias the author does mean here and how this is connected to the minimum separation distance.

line 262: Though you cannot mention it all here, at least some of these previous studies should be mentioned here directly.

Caption Fig. 17: Better to explain here directly what the black dots mean rather than to make a detour via the red curves in Fig. 14.

Fig. 17-19: please see my major comment

Fig. 14-16: I assume the plot is based on data from all performed flight tracks? Actually this is not a valid approach to show the convergence of the measured flux towards its true value. For a given track length the data shows a certain spread, though, strictly speaking the data from the flight tracks is not comparable since the alignment of the flight tracks vary. In an atmosphere with fully isotropic turbulence the approach would be fine, but in an anisotropic boundary layer the scatter among the crosswind tracks might be different compared to the scatter among the parallel-wind tracks, i.e. the root-mean square might depend on the track angle. This should be stated clearly in the text so that readers are aware of this. For example, you can use different colors for the different track angles and provide the the root-mean square values for each track angle separately (e.g. in an additional table), in addition to the "ensemble" root-mean square value you have already given. This might also indicate the optimal track angle with respect to the mean wind direction, supposed the statistics are sufficient.

Fig. 17-19: Is there a difference between parallel and crosswind tracks and any recommendation?

line 274-275: You are right with this when you refer to track length of about 10-30 km. But for even shorter track length the LS86 formula tends to underpredict the error as your data show. Actually, everybody knows that the random error for short track length is remarkably high, but flying only short tracks is sometimes the only way in some situations. I would suggest to be precise when you refer to "short" tracks, the understanding of "short" can be sometimes very different.

line 296: Here I miss some final conclusion. From the results shown, which version of integral length scale calculation does the author recommend?

References:

Stephan R. de Roode; Peter G. Duynkerke; Harm J. J. Jonker, 2004 Large-Eddy Simulation: How Large is Large Enough? J. Atmos. Sci. (2004) 61 (4): 403–421.

---

## Author Comment (AC1) · 18 Dec 2020

First, I would like to extend my thanks to both reviewers for their very thorough and thoughtful reviews. Being a newcomer to this particular area of research, the reviews drew my attention to important gaps in my familiarity with relevant prior work as well as giving me occasion to tighten up and clarify my discussions of both my methods and my findings. The revised manuscript that I will submit shortly will be greatly improved as a result.

In the following, the reviewer's comments are indicated by gray shading. My responses appear in the unshaded space below.

- Grant Petty

**1   General comments**

Based on data from a large-eddy simulation for a stratocumulus topped marine boundary layer, the author performed an ensemble of flight measurements and analyzed the convergence of the sampled fluxes towards their truth, as well as investigated the dependence of the random error on the track length. The author compares the observed random error against the theoretically derived expression by Lenschow and Stankov (1986) and found good agreement for track length of 10-30 km. Further, integral length scales of the turbulent quantities were calculated from the LES data for different flight track angles. The author shows that integral length scales depend on the flight track angle and compares these with the proposed, and still commonly-used, approaches by Mann and Lenschow (1994).

The topic and the content of the paper fits well into the journal and is of high interest to the research community. Due to lack of any alternative ways to estimate the random error for flight measurements, almost everybody uses the expressions proposed by Lenschow and Stankov (1986), though it is already known that especially for shorter tracks the estimated random error often does not reflect the true uncertainty. Here, especially for shorter flight tracks, an improved random error estimation is highly desired in order to avoid misinterpretations of observations.

The paper itself is well written and results are sufficiently presented, though at some points in the text more information needs to be given. At several points in the text it is not clear how variables are calculated and from which data, i.e. from the full 3D LES data or from the sampled space-series along the flight tracks. Also, at some points the discussion is rather short and the findings are not well put into the context of previous research. For example, also the study by Schröter et al. (2000) had already analyzed how the sampled flux converges towards its truth with increasing track length. Even though the atmospheric setups are not comparable, the findings presented here should be put into the context of previous work.

Thank you. I will attempt to add that context.

After extensive review I can recommend the manuscript for publication only after major revisions have been done and extended analysis is presented. My major concern is outlined in the following.

**2 Major comments**

(A) I miss one important aspect in the study. The author shows that the predicted random errors according to the Lenschow and Stankov formula matches the observed standard deviation remarkably well. However, in this study the random error using the LS86 formula is calculated based on integral length scales, correlation coefficients, and fluxes, that were inferred from the three-dimensional LES data. Though it is nice to show that the LS86 formula works well in the theoretical case where all these data is available, this approach does not reflect the reality at all. In reality, the integral length scale need to be calculated based on the flight-sampled data itself, hence, it is exposed to the same sampling errors as the flux is. Particularly for short track lengths, the errors in the integral length scales are supposed to be remarkably high. These biased values then propagate into the LS86 formula, with the consequence that the standard deviation of the predicted random error is also very high, with far reaching implications concerning the interpretation of the measurement, as the random error estimation cannot be trusted anymore for short tracks. In some situations the LS86 formula indicates high random error, but in other situations it indicates low random errors (please see the discussion in Sühring and Raasch (2013) about this). Hence, the comparison of the LS86-predicated error against the truly random error in the way you have done it here is actually not fair and does not help much in the interpretation of observed data. The manuscript would strongly profit if you add such an analysis. I would propose to add following analysis: * How does the integral length scale and the correlation coefficient used in Eq. 5 depend on the track length? Here a direct comparison against the true value is possible since the integral length scales from the 3D LES data are available. * How does the random error behaves when it is calculated directly from the sampled space-series, and how well does it match with the 'true' random error? Based on this, the performance of the LS86 formula can be directly shown for different track length, which would be extremely helpful for researchers.

This is an interesting point. I had previously been assuming that, for a given regime (e.g., neutrally stratified cloud-topped marine boundary layer), integral lengths could be determined and then assumed to be approximately the same in other similar environments. In that case, the domain-averaged ("global") integral lengths determine in this paper might be usable by other investigators, and there would be no need to determine integral length scales on a case-by-case basis. However, you raise the possibility of the latter, so I have added a new section specifically looking at this question.

Surprisingly, especially in light of the findings of Durand et al. (2000), I find that the the standard deviation in the error estimation is not as large as expected and, even more surprisingly, it seems to converge quickly to a value that doesn't depend much on track length. (I also plan add some comments cautioning against interpreting the computed magnitudes of the standard deviation too literally, owing to an apples-to-oranges aspect of the comparison, including with regard to your final comments about cross-wind vs. parallel sampling errors.)

**3  Further comments**

Abstract - Line 2: vertical turbulent fluxes Line 26 - 30: The paragraph is quite imprecise, it is not clear to what refinements done by Mahrt (1998) or findings by Hollinger and Richardson (2005) the author refer to. The introduction would profit if the author would elaborate this a bit more.

I have revised the comment about Mahrt (1998) to say that he "further examined the sampling problem in the context of the problems posed by non-stationarity." I have deleted the reference to Hollinger and Richardson.

As an aside, I will say that as a newcomer to the field of boundary layer meteorology, I'm hesitant to attempt too detailed of a review of prior work on this problem because of the near-certainty that I will miss or misstate something important. My hope is that by providing references to the works that I know about, readers can dive deeper into the subject matter by reference to the original sources rather than depending on my inaccurate interpretation or incomplete knowledge of those earlier works.

line 44: Please remove the word "exceptional". Though it is indeed quite a large setup, it is not exceptional any more. In the last one to two years such setups have become already standard in the LES community.

I have modified the statement. I would be happy to cite other studies using a similarly large setup, but I have not seen them yet.

line 42-44: To the reader of the manuscript it might appear unclear what is meant by this sentence "The single most ...". I suppose the author mean that the smaller scales in the LES are filtered by the subgrid scheme, as well as by numerical errors, being not sampled adequately in an LES.

I have rewritten and restructured much of the introduction in an effort to be clearer.

line 48-49: I disagree with the last part of the sentence. Emulating turbulence measurements in an LES always suffers from the missing subgrid-scale contributions, numerical errors etc., independent which grid resolution is used. Comparing measured turbulence spectra and spectra derived from LES-sampled data will always show a drop-off spectral energy on the smallest spatial scales (about 10 times the grid spacing as a rule of thumb), as it is also the case here (Fig. 2). The relevant question is if, and when, how much the missing subgrid-scale contributions affect the analysis of sampled turbulence data. I would recommend just to rephrase the part with the subgrid-scale flux here.

Your point is well-taken.

end of introduction: I miss a manuscript outline here, to guide the reader through the manuscript.

I have added an outline.

line 79-88: Beyond the fact that a marine boundary layer was simulated, what is the general atmospheric setup? Is this a boundary layer in the trade-wind zones or in a polar region (which latitude). Of course, according to Fig. 1 it becomes clear, but such information should be also given in the text.

I have added some additional information to clarify the meteorological context, which is basically the prevailing summertime close-cell stratocumulus regime off the coast of California.

line 93: How were these spectra calculated? Were they calculated from the emulated flights or from the 3D LES data (1D or 2D spectra)?

I have clarified in the revision that these were calculated from the entire 2-D horizontal domain with subsequent radial averaging.

Fig. 2: Is it temperature or potential temperature? In Fig. 1 profiles of potential temperature are shown, later on the author only refer to temperature, though I cannot find any statement about a transformation.

The transformation is the standard definition based on Poisson's equation. At the low altitudes considered here, the distinction between $T$ and $\theta$, and especially between the perturbations of these quantities, is numerically insignificant.

line 98: With the phrasing "circular eddies" you indirectly imply that the turbulence is isotropic, which isn't the case as plenty of observations and simulation data show. I would recommend to simply remove the word "circular" here.

Removed.

line 99-103: The author describes the spectra sufficiently here, though some references with respect of the minimum domain size of the LES domain are missing. However, I miss some further discussion about possible implications on further results. It is well known that at smaller wavelength the spectral resolution is bounded due to the subgrid-scale model as well as numerical dispersion and dissipation errors, causing these steep drop-off. In most cases this in no big issue as the smaller scales do not contain much energy. But also at the longer wavelengths, the spectral resolultion of LES is bounded by the domain size, especially for humidity (de Roode et al. 2004). From previous studies it is known that structures grow in time, meaning that the spectral peaks move towards larger wavelengths, until the structures cannot grow anymore as they are bounded by the domain size. Somehow the spectra for q indicates this. In case this happens I suppose the integral length scales of humidity are also affected by this, though it won't change much on the overall results I guess. You already bring this up in line 135-137, but maybe it is worth to already bring this up here too.

I have added some more remarks on this issue as well as a reference to de Roode et al.

line 91/104: Though it is only one author, the we-form is repeatedly used throughout the manuscript.

This touches on a controversial subject. Some journals apparently ban the use of first-person ("I" or "we") altogether, requiring the use of the passive voice to describe what was done ("the data were analyzed"). Other authorities rail against the passive voice, urging the use of the active first person when possible. Philosophically, I agree with the second position, but it is rare and distracting to see the singular "I" in journal articles, and it feels egotistical. Even for a single-authored publication, it seems appropriate to use "we" if the reader is being included (e.g., "we are concerned here with X"). Some additional discussion can be found here: https://www.editage.com/insights/is-it-acceptable-to-use-first-person-pronouns-in-scientific-writing

In the end, I have decided to revert to the passive voice except in the "we" case just mentioned.

Fig. 3: Again, is this absolute or potential temperature?

From the purely numerical perspective, it could be either. From a notational perspective, I will fix the inconsistent usage in the original paper.

line 109-11: I disagree with this. The alignment with the mean wind is also visible in q' and T'. w' and q' (or T') correlate actually fairly well as it is typical in a convective boundary layer where the updrafts are created by buoyancy (which in turn is related to q' and T').

My statement was, "This directional anisotropy is far less apparent in $q'$ and is completely absent from $T'$ at low levels." It's admittedly difficult to make objective statements based on a purely visual interpretation of the images, but my statement seems to be consistent with the crosswind and parallel integral length profiles depicted later in Fig. 11 (original numbering, now Fig. XX).

The near-surface correlation between $T'$ or $q'$ and $w'$ is about 0.5, according to Fig. 12 (original numbering, now Fig. XX). I suspect it would be higher still if the directional anisotropy were the same for both variables, but this is admittedly conjecture.

Eq. 1-2 and Fig. 7-8: Here it would be good to already mention that this quantity will be used for the integral length scale calculation. Without this background, which is not clear at this point in the manuscript, this might puzzle some readers. I am not entirely sure what these quantities are actually represent. To my understanding it is the local vertical flux at one point in time which can only be calculated from simulation data in the special case of a horizontally homogeneous boundary layer. To obtain heat fluxes in the traditional sense, time-averaging would need to be applied on top. Hence, w'T' is not a real flux but a quantity used to compute integral length scales. I would recommend to make this clear in the manuscript, i.e. that these temporal local fluxes cannot be compared one to one to flux measurements from aircrafts or towers, neither with respect to the spatial pattern nor with respect to the amplitude. But in the context of the integral length scale calculation such equations and plots does make sense.

I think the reviewer and I see Eqs. 1 and 2 differently and apparently disagree on the second-to-last sentence above. Throughout this paper, we are looking at flux determinations from a single time step of the LES. We are calculating fluxes as though an airplane were flying at a high enough speed that we can invoke Taylor's "frozen turbulence" hypothesis—space can replace time (and vice versa) in the Reynolds averaging of fluxes. So spatial integrals of equations 1 and 2 should be equivalent to temporal averages (e.g., stationary observation point, non-zero wind) and are exactly the fluxes we are purporting to evaluate with our transects through the domain. A spatial integral over the entire horizontal domain gives use the "true" domain-averaged flux, and the linear integrals along the flight tracks give us the imperfect estimates of that "true" flux. In short, we're not just using (1) and (2) to compute integral lengths; we're using them for the actual simulated flux estimates along flight tracks. I did revise the notation, because another reviewer pointed out that is is more common to use $H$ and $LE$ to denote sensible heat flux and latent heat flux, respectively. I also replaced $T'$ with $\theta'$ in (1), because that seems more familiar to some reviewers, even though it makes no significant difference in the numerical results.

That said, I have also been advised by others that I should reduce and consolidate the number of figures, and Figs. 7 and 8 were among those highlighted for deletion.

line 128: one dot too much

Thank you.

line 128-129 and Fig. 9: As the author have already mentioned in the text the momentum-flux profiles look quite noisy which is due to the lack of time averaging. Especially for the momentum flux longer averaging periods are required to obtain smooth profiles. Does the author have access to time-averaged profiles in order to check whether the shown momentum flux profiles changes much? According to the wind profiles shown in Fig. 1, I suppose the momentum flux profiles would decrease linearly with height. Here it would be worth to note this in the text and maybe a statement of how much the comparison against emulated flight measurements will be affected, as the truth in momentum flux is just a first guess.

Again, there may be a fundamental difference of opinion here. I am assuming that time averaging is equivalent to spatial averaging under Taylor's "frozen turbulence" hypothesis in the same way that time averaging *is* effectively spatial averaging when performed by a fast-moving aircraft. I don't think there's anything controversial about that. As for the noisiness of the momentum flux profile, one could just as easily state that that noisiness would be reduced by more spatial averaging (larger domain). My conjecture in this case is that the noisiness of the profile is just the result of the domain not being infinitely large and not in perfect steady state. I don't have access to time-averaged profiles, but I agree that sufficient averaging in time, space, or both would probably result in a linear profile.

In any case, the noisiness (in all three spatial dimensions) of momentum flux is certainly relevant to the flux sampling problem, and it is indeed shown in the original Fig. 16 that sampling errors for this variable are quite large relative to its magnitude.

line 133-134: I am wondering why the lower range for the 90-% contribution changes among the different height levels. To calculate an ogive you actually start at the smallest wavelength (or largest wavenumber) and integrate until the target wavelength. The smallest wavelength, however, is fixed by the cut-off wavelength which depends on the grid spacing, so the intervals should be always something like [2.5m-x1], [2.5m-x2], etc.

Unfortunately, I don't understand the comment. Yes, the ogives are calculated starting at the shortest wavelength (left end of the x-axis), which is why they all start out at zero there. And yes, the spectra are obtained at multiples of the Nyquist wavelength, but that interval is not a visible feature of the plots, so I'm not sure what the reviewer is referring to.

Fig. 11 f-h: It is not entirely clear how the integral length scales for fluxes are obtained. I suppose you calculated $I_f$ from the local fluxes obtained from equation 1-3, but it is nowhere written in the text. Or did you used equation 6-7 to calculate $I_f$?

The integral length scales are calculated using the definition given in Eq. (4). I have slightly expanded the explanation of how this was done. They were calculated for individual variables as well as for for the local flux contributions in (1) and (2) (or, more precisely, the products $w'\theta'$ and $w'q'$, since the constant coefficients play no role).

line 196-201: It would be nice to add the resulting length scales from equation 6-7 into Fig. 11 f-h to have this also visually.

I thought about that when preparing the original figure, but since representative values from those equations can be easily computed by hand using point values selected from Fig. 11 (old numbering), I decided to leave that figure as a depiction only of actual integral scales rather than risking confusing readers with the very imperfect estimates of $I_f$ from (6) or (7). A key finding is that (6) and (7) seem to seriously overestimate the true $I_f$

Section 3.3 and 3.4: The information content is not that big to put it into separate sections, you can easily merge it with section 3.2. But that's of course a matter of taste.

Agree.

line 211: Please add the interval for n directly after the arctan-formula, this will make it easier to understand what is done here.

I have clarified that $n$ is a positive integer.

line 208-215: Before you start with your flight-track setup, you should first start to explain this special problem which is just because you use a stationary frozen flow field rather than a moving framework with advection where this problem will not occur unless the domain size is large enough. This will make it easier for the reader to understand what is done here.

I have expanded the introduction to this section.

Fig. 13: Just a minor point, but it would be nice to mention the flight direction - from southwest to the northeast, or was it the other way around?

Because we are "instantaneously" sampling the entire flight track, there is no definable direction of flight, only a track position and orientation.

line 234-240: I am not entirely sure what is the effect of the insufficiently sampled true value due to lack of temporal averaging, especially for the momentum flux which has been clearly not converged. But as you compare the flight-sampled data against these true flux, and both are calculated based on the same data set, I suppose that this won't have any effect on the following results. But interested readers might ask the same question, so it could be worth to spend one or two sentences here to make this more clear.

This comes back to the question of whether pure spatial sampling is an adequate proxy for temporal sampling (e.g., a fixed tower) or temporal-spatial sampling (e.g., a moving aircraft with finite speed). This entire paper depends on the answer being "yes." I'm not sure how to reintroduce that question at this point without risking confusing readers more than I help them. However, I have tried to highlight the philosophical concerns surrounding this issue a little more clearly earlier in the paper.

As an aside, my impression was that the Taylor "frozen turbulence" hypothesis was originally invoked as a way to justify using time averages at a point as an acceptable substitute for spatial averages, rather than the other way around. Regardless, more averaging in either time or space, or ideally both, will always lead to reduced statistical error, as long as the turbulence statistics are stationary.

244-245: It is unclear which bias the author does mean here and how this is connected to the minimum separation distance.

If you don't enforce a mininum separation between parallel segments of a track, then eventually the entire domain is sampled so thoroughly that the track estimate converges prematurely on the "true" value and the apparent error goes to zero. This is likely to start becoming a problem when the track separation is smaller than the autocorrelation distance (i.e. integral length) of the field being measured, in which the adjacent paths are no longer statistically independent. The bottom line is that once that minimum distance is violated, the determination of flux error starts to be biased low relative to what would observed along a path through a truly infinite domain.

line 262: Though you cannot mention it all here, at least some of these previous studies should be mentioned here directly.

I have added a reference to Grossman (1992). There are probably many other papers with similar findings, but with the deadline to resubmit this revision upon me, I will leave it at that unless more are requested.

Caption Fig. 17: Better to explain here directly what the black dots mean rather than to make a detour via the red curves in Fig. 14.

I have expanded the caption as requested. I have also changed the color to be same red as those other curves so as to eliminate one additional possible source of confusion.

Fig. 17-19: please see my major comment

I have added a new figure and a new section to examine the issue you raised in your major comment.

Fig. 14-16: I assume the plot is based on data from all performed flight tracks? Actually this is not a valid approach to show the convergence of the measured flux towards its true value. For a given track length the data shows a certain spread, though, strictly speaking the data from the flight tracks is not comparable since the alignment of the flight tracks vary. In an atmosphere with fully isotropic turbulence the approach would be fine, but in an anisotropic boundary layer the scatter among the crosswind tracks might be different compared to the scatter among the parallel-wind tracks, i.e. the root mean square might depend on the track angle. This should be stated clearly in the text so that readers are aware of this. For example, you can use different colors for the different track angles and provide the the root-mean square values for each track angle separately (e.g. in an additional table), in addition to the "ensemble" root-mean square value you have already given. This might also indicate the optimal track angle with respect to the mean wind direction, supposed the statistics are sufficient.

This comment is of course correct, but the problem is difficult to satisfactorily address. The creation of periodically continuous tracks allows no choice in their orientation; the angle is $\phi = \arctan(n)$, with $n$ an integer that determines not only the orientiation but also the track length. The 90 degree rotations and mirror images ensure a symmetric distribution about the points of the compass, but it remains the case that all possible orientations relative to the wind direction occur, and few are perfectly aligned either parallel or perpendicular to the mean wind. So any segregation of tracks into two groups will inevitably include many that are neither. I have some thoughts on how to take this into account, but in view of the looming deadline to post a response to the reviews, I will simply have to work them into the revised manuscript rather than addressing them here.

Fig. 17-19: Is there a difference between parallel and crosswind tracks and any recommendation?

There is clearly a large difference in the integral length at lower altitudes, implying that a proportionally shorter track length is needed in the crosswind direction to achieve the same sampling error in the flux measurement. I'm new to this field, but I believe that this is already widely understood.

line 274-275: You are right with this when you refer to track length of about 10-30 km. But for even shorter track length the LS86 formula tends to underpredict the error as your data show. Actually, everybody knows that the random error for short track length is remarkably high, but flying only short tracks is sometimes the only way in some situations. I would suggest to be precise when you refer to "short" tracks, the understanding of "short" can be sometimes very different.

Understood.

line 296: Here I miss some final conclusion. From the results shown, which version of integral length scale calculation does the author recommend?

If I'm understanding the question, this is apparently about whether (6) and (7) are adequate substitutes for the directly determined integral length from (4), when the latter is not available. It appears from the results presented that (6) and (7) are prone to overestimate the integral length and thus overestimate the sampling error.

---

## Author Comment (AC2) · 18 Dec 2020

**1 General comments**

First, I would like to extend my sincere thanks to both reviewers for their very thorough and thoughtful reviews. Being a newcomer to this particular area of research, the reviews drew my attention to important gaps in my familiarity with relevant prior work as well as giving me occasion to tighten up and clarify my discussions of both my methods and my findings. The revised manuscript that I will submit shortly will be greatly improved as a result.

In the following, the reviewer's comments are indicated by gray shading. My responses appear in the unshaded space below.

- Grant Petty

> Regarding the scope of AMT scientific questions, the question of the publication of this study in this journal may be raised. Indeed, even if the topic is about airborne flux measurements, the study is based exclusively on results from numerical simulations. It is regrettable that no observations are used in this study, either to be confronted with the simulation output or to apply the results obtained, for example on past measurement campaigns.

According to the AMT's statement of scope, "Papers submitted to AMT must contain atmospheric measurements, laboratory measurements relevant for atmospheric science, **and/or theoretical calculations of measurements simulations with detailed error analysis** including instrument simulations." I relied on the highlighted phrase when deciding whether to submit to AMT. The other reviewer seemed to agree that it was appropriate, stating "The topic and the content of the paper fits well into the journal..."

> The track definition used by the author can lead to flight tracks greater than the domain size thank to the cyclic boundary conditions of the LES. Nevertheless, as mentioned by the author, the finite LES domain is not able to reproduce structures greater than the domain size. Is a domain of 5.12 x 5.12 km2 is therefore large enough to study airborne sampling and eddy correlation flux estimation? With a larger domain size, the characteristics of the simulated turbulent structures may be different. With this issue of the limited size of the domain, does a LES with a larger mesh grid and a larger domain have been appropriate? Taking the example of the University of Wyoming KingAir aircraft mentioned by the author, with a true air speed of 85 m/s and a measurement frequency of 25 Hz, the sampling spatial resolution is then about 3.5m. Thus, a grid mesh 3 times larger than the one used here could be adequate.

The important point here is that I had no control over the parameters of the simulation. I didn't run the LES but rather requested the output from the model run previously published by Matheou (2018), and he was kind enough to provide it.

I agree that the domain dimensions aren't as large as would be ideal for this kind of study, but this particular simulation spanned a larger-than-average range of scales and seemed likely to be a better-than-average representation of turbulent fields at very low altitudes (e.g., 10 m or 40 m). My interest in undertaking this analysis was motivated in part by my involvement with very low- and slow-flying aircraft—e.g., $\sim$10 m altitude and 20 m/sec. Particularly at the lowest altitudes, I'm less concerned about the limitations imposed by the 5 km domain size. If we imagine that the goal is to accurately sample the near-surface fluxes within the limited domain, via repeated parallel passes (but without the complication of turning a physical aircraft), then what happens outside the domain seems less important.

There are many figures (19 in total), some of which seem redundant or could be concatenated. Several of them are simply mentioned in the text without being analyzed or discussed. The question of the relevance of these figures may arise, not helping to clarify the main message of the article. It obviously seems appropriate and necessary to present the simulation with the help of a few figures, however, it is only from figure n∘13 that the central purpose of the paper begins to be addressed.

I have consolidated several figures into single figures and deleted three others. There are now 11 rather than 19 separate figures.

**2 Specific comments**

**2.1 Introduction**

The works of Lenschow and Stankov (1986), Lenschow et al. (1994), and Mann and Lenschow (1994) were not only based on theoretical considerations and statistical models but also on observations. It might be useful to include in the introduction, some studies on experimental data and field campaign. In general, the introduction could be enhanced in terms of bibliographic references, such as Brooks and Rogers (1997) Cook and Renfrew (2015) or Brilouet et al. (2017).

I will add those references and try to add appropriate context. That said, as a complete newcomer to boundary layer meteorology, there is a danger that I will mistate or misinterpret something important, so my inclination is to let the cited references speak for themselves as much as possible.

Line 34: The LES is able to resolve explicitly the major part of the turbulence but it remains a sub-grid contribution. Even if with a 1.25m resolution, this contribution becomes rapidly negligible with the altitude, it might be useful to mention that total turbulence = explicitly resolved + subgrid contribution.

I now extend the sentence in question to say, "leaving only a small fraction of the total turbulent exchange to subgrid-scale parameterizations, especially at levels much above the surface."

After line 42, it is not clear if we are still in the introduction section or if the section "description of the method" has already started. It would be useful if the main goal of the study could be more clearly highlighted and if an outline of the article were provided at the end of the introduction before going into the details of the simulation and the method.

I have significantly reorganized the introduction and data sections, including the addition of an outline of the article.

Line 60: It is correct that using a LES to examine the aircraft flux sampling problem in MABL is unique. Nevertheless, it can be mentioned that previous studies compared LES outputs with airborne measurements such as Brilouet et al. (2020) even if the resolution was coarser.

I have added a mention of that paper a bit earlier in the introduction.

**2.2   Data**

The case study is from the field campaign DYCOMS-II, Are there any observations that might be relevant to the study?

A variety of airborne measurements were taken, as described in part by Stevens et al. (2003). I have not attempted to acquire these measurements or to independently validate the LES, which would be a major effort in its own right. Some discussion of the realism of the LES results is given by the creator of the LES model in Matheou (2018).

The case study is a nocturnal cloud-topped marine boundary layer. When the author describes the environment, a few elements describing the main characteristics of this type of stratocumulus condition could be instructive for the reader (such as $z_i$ at the cloud top, the strong inversion with entrainment at the cloud top, . . . ).

Since the top of the cloud layer coincides with the inversion———and thus the top of the boundary layer———at about 840 m (the maximum height for any non-zero cloud water anywhere in the domain is 885 m), $z_i$ at cloud top is basically 1. The strong inversion with cloud top entrainment is characteristic of marine stratocumulus clouds in this region. However, because my analysis in the paper is focused entirely on the clear-air portion of the boundary layer well below cloud base, I have chosen to omit details concerning in-cloud or cloud-top processes.

Figure 2: the figure is rather small. The units of the power spectra are not mentioned. Does it might be interesting to present normalized spectra (by the variance: $kF(k)/\sigma_X^2$)? Does the spatial wavelength is $\lambda = 1/k$ or $\lambda = 2\pi/k$?

I apologize for the small figure. I had accepted the default scale parameter provided in the AMT style template but should have increased it.

The power spectrum is initially computed as a function of inverse wavelength, meaning the wavelength $\lambda$ of a complete cycle. As indicated in the axis labels, I transformed the spectra in the plots so that the horizontal axis is $\lambda$ rather than spatial frequency. The amplitude plotted as $kF(k)$ without normalization, so the units should be variance per unit $\log(\lambda)$, with the units of the variance depending on the variable plotted. I did not normalize; doing so has no effect on the curves other than shifting them vertically. The focus in these plots is on the slope. I later realized that I omitted the conventional factor of $2\pi$, so I will correct the notation on plots to make sure it is consistent with common usage.

Line 95: It might be interesting to compare with previous works.

Again as one very new to this subject area, I am not sufficiently familiar with prior work looking at wavelengths of peak energy in turbulence to be able to quickly identify the most relevant studies. I apologize. I hope that others who do have that familiarity will be able to interpret my results within the context of previous findings and/or suggest studies that should be cited.

Lines 96-98: At 40m height (0.05 zi), this is the surface layer. How much the turbulence is explicitly resolved at this height? What is the vertical profile of TKE resolved / total TKE? Also, the surface layer may have different characteristics than the layer above. Does the Monin-Obhukhov Similarity theory (MOST) is available? It would be interesting to enhance the discussion with some references on the turbulent structure inside the surface layer such as Katul et al. (2011) or Sun et al. (2016).

The output file I received from Dr. Matheou does not include the parameterized subgrid components of the TKE or fluxes, so I can't directly answer that reasonable first question. However, I understand from casual conversations, possibly incorrectly of course, that the resolved component of the turbulence should dominate once you're much above 5 or so grid levels, where $\Delta z = 1.25$ m. The flux sampling error analysis depends only on realistic spatial statistics of the turbulence on the scales containing most of the energy. The subgrid component of the turbulence would likely correspond to a more or less linear extrapolation of the plotted spectra below approximately 10 m wavelength. As a fraction of the total variance, that extropolation doesn't add much, and I don't think it would much change the general findings in this paper either, especially at 40 m and higher.

Regarding turbulent structure of the surface layer, I'll admit that I'm not a boundary layer theoretician, so I wouldn't be qualified to undertake that discussion.

Lines 99-100: Do the spectra of temperature and specific humidity reveal more energy at longer wavelength due to the influence of mesoscale on those parameters? If the domain was larger, would the wavelengths be longer?

Satellite images clearly reveal variations in stratocumulus cloud decks on larger scales, but I don't know what fraction of the total variance in velocity, temperature, or humidity is represented by those longer wavelengths. The safest way to interpret the present paper is as simulating airborne flux measurements over a restricted 5 km domain, so that things happening beyond the lateral boundaries aren't really relevant to the basic question under consideration.

Line 101: What is the reason that the horizontal wind speed spectrum has no significant dependence on height?

Again, I'm not a boundary layer theoretician, so I can't be sure of the answer. My suspicion is that because horizontal flow isn't obstructed by the upper or lower boundaries, and because there is relatively little friction near the ocean surface, vertical mixing within this nearly neutrally stratified BL is efficient enough to maintain a fairly homogeneous spectrum.

Lines 104-105: The author has chosen four representative heights, one at 10 m and another at 40 m. Are these heights characteristic of airborne measurements?

In the paper, I now mention examples of airborne measurements at 40 m (Cook and Renfrew 2015) and at 100 m and 400 m (Desai et al. 2020, in press). 10 m is also of interest to me in light of possible future turbulence measurements from a very low-flying ultralight airplane or drone.

Figures 3-5: 3 figures are considered for 4 lines. It would be interesting to concatenate them into a single figure. It will be easier to compare the characteristics of each parameters and their evolution with the height (for example with left panels at 10 m, middle panels at 40 m and 100 m and right panels at 400 m with a parameter by row).

This is a good suggestion, and I have merged them into a single figure with rows corresponding to height and columns corresponding to variables.

Lines 106-110: Also, a link with previous work would be valuable.

While the results I describe seem consistent with my expectations, I realize now that I cannot point to a specific study describing smaller-scale turbulent structures in the neutrally stratified marine boundary layer. I would be happy to add any that are suggested to me. In the meantime, I have removed the word "expected" from my description.

Figure 6: Is this figure really essential to the article?

I have deleted that figure.

Line 114: It might be helpful to define the sensible (H) and latent (E) heat fluxes. Commonly, the E notation refers to the surface moisture flux or evaporation ($E = \rho \times w'q'$). Perhaps the LE or LvE notation is more appropriate for the latent heat flux.

I have changed the notation to $LE$.

Lines 114-119: Is the definition of sensible and latent heat fluxes and their expressions as a function of fluctuations valid at different altitudes in the boundary layer? Is it not defined only for surface exchanges? The sensible heat flux is the amount of heat exchanged between the surface and the atmosphere and the latent heat flux represents the energy released or absorbed during a phase change. I may be mistaken and in that case, I apologize for this unwelcome comment.

The actual exchange of both heat and latent heat *at the surface* is of course a non-turbulent (diffusive, molecular) process. Turbulent transport within the atmospheric becomes the dominant process once you get a short distance above the surface. The eddy covariance method is inherently a measurement of vertical turbulent flux through a plane *at the level of the instrument*, whether close to the surface or higher up. Depending on stratification, storage, etc., it may or may not be an adequate representation of the surface flux.

Figures 7 and 8: These figures are not described or analyzed in the article. Are they essential to the article?

I have deleted those figures.

Line 130: It would be interesting to explain the TKE profile and how this is expected, in terms of the processes involved, given the case study under consideration. Here again, a connection with previous studies on this subject would be appreciated.

Once again, I'm not a boundary layer theoretician, and I'll admit to not being familiar with previous studies on that particular subject. That said, it seems to me that with viscous dissipation of TKE occurring only at the smallest scales (and therefore presumably being relatively slow), and with this neutrally stratified boundary layer readily mixing in the vertical, it might make sense that TKE would be almost uniformly distributed through that depth once you get well below the radiative forcing at the top of the cloud layer.

**2.3 Integral length scales**

In this section, the work of Lumley and Ponofsky (1964) could enhance the bibliography as a pioneer on these issues.

While I have this book in my private list for future reading, I haven't ever held a copy in my hand, and I'm currently unable to access a library copy due to the COVID shutdown. I hesitate to cite a source without stating what I'm specifically citing it for. My apologies.

Line 140: It is the first time, since the introduction, that the random error is mentioned. As this is the main focus of the article, wouldn't it be a good idea to highlight it further? The current design of the article suggests that it is secondary to the integral scales.

I will expand the discussion of the random error and introduce parts of that discussion earlier in the manuscript.

Line 149-150: To introduce the random error in a simplified point of view, is the equation 1 of Lenschow and Stankov can be relevant?

As a definition, yes. See above.

The spatial correlation $\rho_{w\psi}$ is defined twice (line 156 and line 164).

Thank you.

Line 158: In order to specify the experimental difficulties in estimating the integral length scale, the study of Durand et al. (2000) could be instructive.

I have added that reference.

Figure 12: Even if the random error definition contains the correlation $\rho_{w\psi}$ is the figure really essential to the article? Simulated aircraft measurements

I think it's worthwhile to show these since the values are utilized in the error determinations. In particular, I make reference in a couple of places to the fact that $\rho_{w,U}$ crosses zero near 400 m to explain why momentum flux is small there and the error factor becomes very large.

Lines 208-209: This sentence perfectly summarizes the main topic of the study. Isn't it a bit late? This message does not appear clearly enough throughout the article.

In the revision, I have tried to make that clearer in the introduction.

Lines 246-247: As mentioned in the general comments, I have some concerns about the domain size with respect to the characteristic scales of fluxes that can be observed during airborne measurement campaigns. Consequently, the results that will arise from this study seem difficult to be transposed to measurement campaigns.

Yes, the domain size is a limitation. More nearly ideal would be a 50 km domain with 1 meter resolution, but that is not currently feasible, and I am in any case working here with someone else's LES output. I do not currently have a better source of model output for this particular kind of analysis, but I believe the analysis offered here is a small step in the right direction, provided that the unavoidable limitations, especially with respect to large-scale contributions to fluxes, are kept in mind.

Line 249: Another way to check Taylor's hypothesis, for airborne measurements, the true air speed (here V = 85 m/s) can also be compared to the intensity of the turbulence $(\overline{u'^2})^{1/2}$. If $V \gg (\overline{u'^2})^{1/2}$ then the statistical properties of the turbulence field are assumed to be unchanged over the considered time interval.

This may be true, but I haven't heard it before, and I don't know whom I could cite as a source for that relationship. I will continue thinking about why this statement is valid, as I'm not immediately seeing the reason. If true, then we're talking about the standard deviation of $u'$, which was shown in Fig. 6c (now deleted in the revised manuscript) to range from around 0.5 to 1.0 m/sec.

**2.4   Results**

Figures 14-16: These three figures could be concatenated into one. Moreover, even if these figures are at the core of the study, they are barely detailed and analyzed (Figure 15 is barely mentioned).

I agree with this suggestion and have combined them into one figure.

Line 261: Including bibliographic references would be valuable.

I will look for appropriate references.

Figures 17-19: In order to facilitate the understanding of the figures, it can be useful to keep the empirical RMS error in red rather than changing the color. Are the parameters in blue necessary? If so, would it be better to include them in a table? As the minimum track length L10 for 10% relative accuracy is one of the main results, would it be a useful to group them together, for each flux and each altitude, in a table?

Again good suggestions. I have combined the three figures into a single figure, changed the colors, and moved the parameter and L10 values to tables.